# Conditioned haptic perception for 3D localization of nodules in soft tissue palpation with a variable stiffness probe

Nicolas Herzig[1]*, Liang He[2], Perla Maiolino[3], Sara-Adela Abad[4,5], Thrishantha Nanayakkara[2]

**1** Department of Automatic Control and Systems Engineering, University of Sheffield, Sheffield, United Kingdom, **2** Dyson School of Design Engineering, Imperial College London, London, United Kingdom, **3** Oxford Robotics Institute, University of Oxford, Oxford, United Kingdom, **4** Department of Mechanical Engineering, University College London, London, United Kingdom, **5** Institute for Applied Sustainability Research, Quito, Ecuador

* n.herzig@sheffield.ac.uk

**Data Availability Statement:** The dataset supporting this article is available on the ORDA (Online Research Data) database (provided by figshare), DOI: 10.15131/shef.data.12732824.

## Abstract

This paper provides a solution for fast haptic information gain during soft tissue palpation using a Variable Lever Mechanism (VLM) probe. More specifically, we investigate the impact of stiffness variation of the probe to condition likelihood functions of the kinesthetic force and tactile sensors measurements during a palpation task for two sweeping directions. Using knowledge obtained from past probing trials or Finite Element (FE) simulations, we implemented this likelihood conditioning in an autonomous palpation control strategy. Based on a recursive Bayesian inferencing framework, this new control strategy adapts the sweeping direction and the stiffness of the probe to detect abnormal stiff inclusions in soft tissues. This original control strategy for compliant palpation probes shows a sub-millimeter accuracy for the 3D localization of the nodules in a soft tissue phantom as well as a 100% reliability detecting the existence of nodules in a soft phantom.

## Introduction

During the last decade, the human-robot or robot environment interactions have been one of the main foci of research in robotics. Advances in tactile sensing [1, 2], compliant robotics [3, 4], soft robotics [5, 6] and new control methods [7], have been improving robot capability to interact with the environment. This trend is one of the promising advances that can bring new opportunities for robotic applications in the healthcare field. The possible outcomes of human-robot interaction in medical application range from robot-assisted medical imagery [8, 9], teleoperated surgery [10], medical training [11, 12] to robotic-assisted examination. This paper focuses on medical palpation of soft tissue to localize hard nodules.

Robot-assisted palpation aims to use a robot to perform haptic examinations of a patient in place of a medical practitioner. Often used as an early-stage examination, the goal of this haptic investigation is to estimate the mechanical properties such as the texture, the stiffness, or the

**Funding:** NH PM and TN were funded by the United Kingdom Engineering and Physical Sciences Research Council (EPSRC) MOTION grant [EP/N03211X/2]. TN was also funded by the EPSRC RoboPatient grant [EP/T00603X/1]. https://epsrc.ukri.org/ The funders had no role in study design, data collection and analysis, decision to publish, or preparation of the manuscript.

**Competing interests:** The authors have declared that no competing interests exist.

consistency of an organ. It can also help to track the size and the position of a nodule or a tumor in soft tissues [13].

In such physical examinations, the compliance of the robotic probe plays an important role due to several reasons such as safety and stability under tissue uncertainty. Since the compliance is defined as the inverse of the stiffness, we mean by compliant system a physical system with low stiffness. By opposition, a stiff system is a system with low compliance. We will deliberately use the two words compliance and stiffness inconsistently in this paper since some ideas are more intuitive when expressed using the stiffness, while some others are more intuitive with the compliance. However, a stiff robotic probe to explore patient tissues relies only on compliance of the tissues which increases the risks of hurting the patient. Our previous study has shown that the compliance of the probe can be used to maximize haptic information gain during hard nodule depth estimation in soft tissue exploration [14]. There have been several approaches to give compliance to robots. These approaches can be based on mechanisms such as Variable Stiffness Actuators (VSA) [15, 16], passive elements integration [17], or based on control algorithms such as impedance controllers [18, 19]. The role of the compliance and the impedance have been widely studied for stability analysis, disturbance rejection or energy consumption, but the impact of such compliance on the perception remains an open question. Thus, the mechanical impedance of a system can, for instance, be used to filter the information and signals measured by a robot while interacting with the environment [20].

In the past, most approaches to robotics treated perception (sensing) and action (actuation) as decoupled phenomena. It seems clear that in the context of haptic exploration or robotic interaction with an uncertain environment, actions taken by the robot directly affect haptic perception. Our approach presented in this paper is inspired by recent studies that provide a better insight into the interaction between perception and action [14, 21]. In particular, we propose a novel control algorithm tested with the Variable Lever Mechanism (VLM) probe [22] to detect and localize an embedded nodule in soft tissues. Thanks to its controllable stiffness and its two sensing modalities (kinesthetic and haptic) we used the VLM probe to demonstrate that the compliance of the joints plays a significant role in the haptic detection, the localization, and the depth estimation of a nodule. The proposed algorithm uses the compliance of the VLM probe to perform either a local (longitudinal sweep) investigation using only the tip-end of the probe or a wide (lateral sweep) investigation using the palmar region of the probe's tip and its tactile sensor. The stiffness is tuned to maximize the information gain (entropy reduction in a random variable) during the explorations by using Bayes inference and likelihood functions. These likelihood functions are built from a prior knowledge obtained during previous palpation or thanks to Finite Element (FE) simulation.

The remainder of the paper is organized as follows. In the next section, the previous research done on robot-assisted palpation are presented. We will give an insight into the research we accomplished on variable stiffness palpation along with other approaches addressed in the literature. Then, we describe the methods used to study the effect of the stiffness of the VLM probe on the perception and detection of a hard nodule embedded in a soft tissue phantom. In particular, we detail the VLM probe's hardware, the phantom characteristics, the lateral and longitudinal sweep experiments, and the Finite Element (FE) simulation. The next section discusses the experimental and simulation results we obtained. The following section presents the likelihood functions obtained experimentally and from the FE simulations. The new algorithm to condition haptic perception and localize stiff inclusion in soft tissues is then detailed and tested in another section. Finally, we discuss and conclude about the performance of the algorithm and the impact of the stiffness variation on perception tasks.

## Related work

During the past two decades, we have witnessed an increasing trend to use Minimally Invasive Surgery (MIS) procedures over open surgeries. Even if we can find some exceptions such as robotic probes to measure the blood flow [23], most of the studies on robotically assisted palpation have been motivated to give local information during MIS that the surgeon would have obtained by manual palpation in open surgeries.

The majority of robotic palpation probes aims to detect hard inclusions in soft tissues. In particular, these probes are mainly designed to detect tumors or anomalies which are generally stiffer than the surrounding healthy tissue. The main types of sensors used to do so are the kinesthetic sensors and tactile sensors. In this paper, we refer to kinesthetic sensors, the sensors that aim to give a signal related to the force or the torque applied at a probe joint level. On the other hand, the tactile sensing is referring to fingertip contact sensing using taxel images. Tactile sensing is usually representing the behavior of the mechanoreceptor at the skin level. Finally, It should be noticed that in this section we omit work on medical imagery such as x-ray, CT scans, or ultrasound which are also solutions to detect hard inclusions but not based on haptic palpation.

Kinesthetic palpation probes measure the force and/or torque during the tissue indentation by the probe. For instance, Ahn *et al.* [24] developed a force sensing probe for prostate cancer detection and tested it on ex-vivo prostate tissue samples. They concluded on the interest of studying the tissue elasticity to estimate the presence of a nodule. Thus, they gave the likelihood of different elasticity levels in healthy and cancerous tissues respectively with the average Young's modulus they measured as well as the standard deviation. Liu *et al.* [25] and Sangpradit *et al.* [26] have developed a rolling indentation probe to detect different abnormalities embedded in porcine kidney samples. By computing a reaction force map after probing, they studied the role of indentation and nodule depth and size on the force measurement. They have shown in particular that the deeper and the smaller the nodule is, the hardest it is to detect it.

Tactile sensing is based on the measure of the local contact pressure between the probe and the tissue. Generally, the information given by these sensors is an array of values taken at spatially distributed measurement points also called taxels. Several technologies have been used to integrate tactile sensing in robotic palpation probes. For instance, Kwon *et al.* [27] designed a tactile sensor for robotic palpation based on an array of pressure-sensitive resistors. Xie *et al.* [28] used an array of optical sensors to measure the contact pressure during MIS palpation.

Trejos *et al.* [29] used a combination of tactile sensors and a kinesthetic sensor. The aim of the tactile sensor is to give a pressure map of the palpated region where the kinesthetic sensor is used only to control the robotic arm where the probe is attached to. They also mentioned using a hybrid impedance controller optimized for a stiff arm with kinematic redundancy. However, they did not investigate the impact of compliance during palpation. In our work, we use tactile and kinesthetic sensing in succession and use probe stiffness control to improve haptic perception efficacy during palpation.

Most of the previous work is based on a stiffness evaluation of the tissues and performed with a rigid probe. The interest of using a compliant probe is not only to avoid relying on the tissue compliance for safe robot-tissue interactions, but also to increase the robustness against misalignment of the probe with the tissue. For example, the probes designed by Jia *et al.* [30] or Faragasso *et al.* [31] are based on passive serial elastic components. A soft robotic approach has been followed by Pacchierotti *et al.* [32] using the BioTac sensor that is based on the measurement of the deformation and internal fluid pressure of a compliant fingertip.

Palpation behavior also affects the quality of haptic perception [14, 33]. Several palpation strategies have been proposed in the literature, but most of them have been developed for stiff palpation probes. Therefore, these strategies do not include stiffness control. A common strategy involves probing point by point to identify the local stiffness and refresh a stiffness map each time a new point is available. It is the strategy used by Ayvali *et al.* [34] who used a Bayesian optimization to determine where the next palpation point should be to reduce uncertainties. Garg *et al.* [35] also used a point by point method with a Gaussian process adaptive sampling in order to estimate the shape of the stiff inclusion efficiently. Using point by point strategy, Hoshi *et al.* [36] proposed an algorithm to optimize the stiffness estimation of the palpated tissues by coupling the force measurement with a predictive model based on the Finite Element Method. Nichols *et al.* [37, 38] have developed a point by point palpation strategy which is based on machine learning trained with ultrasonic elastography image. This algorithm is able to segment hard inclusion in soft tissues by giving the shape of the hard inclusion and an estimation of the local stiffness. Park *et al.* [39] presented the results of a similar machine learning control strategy to segment real cancerous breast samples with a micrometer scale probe. The point by point palpation strategy is really efficient for segmenting hard inclusions or detecting the shape of these abnormalities, but the uncertainties decrease with the number of points and it is often needed to use many points to obtain an accurate estimation. Multiplying the number of points also means increasing the examination time of the tissue, which can be inconvenient for the patient during an in-vivo examination.

Another approach to detect and diagnose hard inclusions in soft tissue is sweeping the probe on soft tissues. This method gives spatially continuous information of the palpated tissue, which reduces the time of examination, but the measured data often suffer from dynamic disturbance and non-linear effects due to the viscoelasticity of the tissues. Chalasani *et al.* [40] proposed a palpation strategy based on sweeping with a sinusoidal normal force profile. Since only the point where the force is maximum are used for the estimation of the hard inclusion, this strategy can be considered as a hybrid method between the sweeping method and point by point method. Salman *et al.* proposed an algorithm for a stiff probe that computes an optimal trajectory of sweeping palpation after based on prior knowledge obtained with point by point palpation [41]. They, in particular, have shown that switching from point by point to sweeping can save examination time. Ahn *et al.* [42, 43] have developed a probe for prostate cancer diagnostics. This probe performs a rotational sweep and measures the force during the sweep. The force is then compared to experimental and FE simulations to diagnose if the tissue contains a tumor or not. Based on the study of how humans perform palpation exploration, Konstantinova *et al.* [44] proposed an autonomous probing strategy following an auto-regressive force regulation to estimate the depth of a nodule in soft tissues. Nevertheless, all these sweeping strategies have been developed for stiff probes.

Our previous studies [14, 45–47] were also inspired by how humans regulate stiffness of fingers during soft tissue palpation. In the previous study [47], we have shown that the stiffness of the arm and hand joints is modified during the longitudinal sweeping exploration of soft tissues by varying the level of co-contraction of antagonistic muscles. The previous strategies to control the probe's stiffness was based on Markov chains and shown a reduction of the number of sweeps (5 Bayesian iterations to reach 80% of confidence) needed to estimate the depth of a nodule compared to a strategy with a random stiffness. The new algorithm proposed in this paper allows not only to estimate the depth of the nodule but also to perform the full 3D localization. Contrary to the previous algorithm, the proposed strategy conditioned the likelihood of the force peak prominence to minimize its variance and maximize the haptic information gain. The compliance of the probe is used to switch from lateral sweep palpation, which provides information for a wide area, to longitudinal sweep palpation, which investigates tissue

along a straight line passing over the suspected location of the nodule, without reorienting the probe. Finally, with this new strategy which combines tactile and kinesthetic measurements with likelihood conditioning, we increased the size of the probed area, we added the localization of the nodule, we improved the resolution of the depth estimation from 3mm (in [47]) to 0.2mm without increasing the number of Bayesian iterations (on average).

## Materials and methods

To study the role of the joints' stiffness on haptic perception during 3D localization of nodules in soft tissue palpation, we first describe the experimental setups and protocols. In the following subsections, the VLM probe hardware and phantom fabrication are presented. We then describe the sweeping directions studied in this paper. Finally, the FE simulation used to model the behavior of the probe during longitudinal sweep is detailed.

### Variable stiffness palpation: The VLM probe

As described in [22], the VLM probe (see Fig 1) is composed of a variable stiffness joint based on a variable lever mechanism and two sensors: a *Cyskin* tactile sensor [2] placed on the VLM probe fingertip and an *ATI NANO 25* placed at the equivalent wrist level. The stiffness variation of the VLM probe is based on position control of a flexible carbon rod. Changing the position of this carbon rod modifies the active length (the part which can bend) of the rod and, by cantilever effect, it changes the stiffness of the joint. An analytical model to correlate the carbon rod position to the equivalent angular stiffness of the joint has been given in our previous study.

Fig 1 shows the design of the VLM probe. The VLM probe is based on a revolute variable stiffness joint composed of 2 rigid links (the base link and the tip link) connected with a revolute joint in parallel with a deformable carbon rod. This carbon rod acts as a variable spring that allows the stiffness of the joint to be controlled thanks to an *Actuonix L12-30-50-6-I* linear actuator. This actuator slides the carbon rod through the base link and the tip link changing the length of the carbon rod that can be bent (active length). As one can see, the hole in the base link has been designed such as that the carbon rod can slide axially but is constrained radially to prevent bending of the rod in the base link. On the other hand, the hole in the tip link is large enough to allow the carbon rod to bend in. A PTFE cylinder is used to transmit the radial forces between the tip link to the carbon rod. This PTFE cylinder has been designed

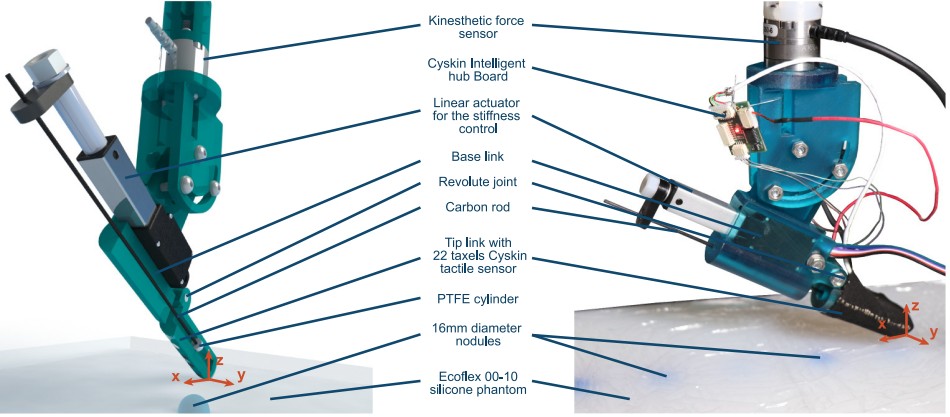

**Fig 1. VLM probe and phantom.** Left: illustration showing the Variable lever mechanism of the VLM probe with a mid-sagittal cut of the tip link and base link. Right: VLM probe and phantom with nodules with different depths.

to slide easily axially when the actuator is translating the carbon rod. Adjusting the active length of the carbon rod changes, by cantilever effect, the amount of force required to bend the rod and by consequence the angular stiffness of the probe.

In order to describe the movement of the probe, we need to define a frame. First, we define the axis **z** as the direction of the normal to the phantom surface. We then define the **x** axis as the intersection between the tangent surface of the phantom and the mid-sagittal plane of the probe. Finally, the **y** is defined in order to obtain a direct orthonormal frame (**x,y,z**). In the rest of the paper, this reference frame will be used to describe the directions of forces or displacement.

To move the VLM probe on the phantoms, the probe is attached to a 3 axis Cartesian robot. This robot is composed of an *Aerotech ANT130-XY* stage and an *Actuonix L16-50-150-12-P* linear actuator which allows the probe to be moved in the horizontal plane (**x**, **y**) and vertically (**z**) respectively. Two *National Instruments* cards are used to acquire the sensors' signals and control the vertical position and the stiffness of the probe. Especially, a *NI PCIe-6320* is used in order to acquire the force sensor signals whereas a *NI USB-6341* controls the two linear actuators positions. The programs to run the experiment and the algorithm have been implemented using C++.

In this paper, the indentation refers to vertical displacement of the actuator of the VLM probe along negative direction of **z** (positive when going down) instead of the depth of the probe's tip in the phantom. Additionally, the indentation 0mm refers to the point where the contact between the probe and the phantom starts. To deal with the issue of alignment between the phantom and the XY stage, the surface roughness of the phantom and possible deformation of the latter, the tangent surface of the phantom (0mm indentation position) is autonomously redefined each time the region of exploration is changed.

To autonomously detect the 0mm indentation position, the method is based on an indentation (without sweeping) and the detection of variation in the kinesthetic sensor measurement. This detection strategy aims to improve the robustness of the nodule detection by improving the accuracy of the indentation measurement. This is particularly important since related studies have shown that a variation of indentation can impact the nodule depth estimation [25, 47].

For all the experiments, the Cartesian coordinates (*x*, *y*, and indentation) of the probe are measured at 50Hz for *x*, *y*, and 10kHz for the indentation respectively. The forces and the position of the actuator which control the stiffness are also acquired at 10kHz. Finally, the tactile pressure is acquired at 20Hz. The acquisition rates are different, but all the signals are time-synchronized.

The angle between the surface of the phantom and the probe is set at 32˚. This angle has been chosen to optimize the number of taxel in contact after indentation during the sweeps.

## Soft tissue phantom with nodules

The phantom is made of platinum-catalyzed silicone Ecoflex 00-10 (Smooth-On, Inc, USA), with 6 nodules embedded at a depth of $2 \times 2$mm, $2 \times 4$mm, $1 \times 6$mm and $1 \times 8$mm (the two additional nodules 2 and 4mm deep have not been used as prior knowledge in the likelihood functions but are used to test the proposed detection algorithm.). It has to be noticed that we define as nodule depth the distance between the surface of the phantom and the upper tangent plane to the nodule. All nodules are made of acrylic with a diameter of 16mm. The phantom is cast at room temperature with a width of 150mm and a thickness of 25mm while the distance between each nodule is 40mm. The ratio between the tissue thickness and the nodule diameter has been chosen accordingly to the one used in related studies in the literature [25, 29, 47].

The mechanical properties of the materials and components of the phantom are described in the Finite Element Simulation subsection.

In the presented study, we limited the nodule depth to 8mm to reduce the average palpation force level and minimize damage to the probe. Indeed, as shown in related works, detecting deeper nodules requires deeper indentation and by consequence, higher forces. Higher forces also lead to faster degradation of the tissue in repeated trials, making it difficult to compare results. Moreover, the lateral sweeps used an array of capacitive tactile sensors, that saturates if an excessive force is applied to locate the nodule. To avoid saturating the tactile sensor, we used a nodule depth that meets all hardware requirements to demonstrate the role of stiffness variation in conditioning the haptic perception during 3D localization of nodules in soft tissues.

These materials have been widely used to simulate human soft tissues mechanical properties. In particular, Ecoflex 0010 has been used in biomedical simulators to practice abdominal palpation [11] or needle insertion [48]. The size of the nodule represents the size of a tumor of type T1 ($<$2cm) for the breast or liver cancers. This is, according to the TNM classification, the earliest stage where the nodules can be detected by palpation [49].

Ecoflex 00-10 is a rubber silicone that has a high coefficient of friction with the probe which affects the motion of the probe. For this reason and to protect the phantom, we wrapped the phantom in an additional layer of plastic film. It has to be noticed that, except for the friction coefficient, this layer of plastic film is not taken into account in the simulation.

## Sweeping directions

In this study, we aim to understand the role of the stiffness on the detection and 3D localization of a hard nodule embedded in soft tissues. Two types of sweeping with different aims are presented in this paper. These two palpation directions show distinct results that have been exploited in the Bayesian algorithm for nodule detection presented in a further section of this paper.

The aim of the two sweeping directions is to reproduce some human participants' palpation strategies that we observed during our previous study [44]. We have shown in this study that the palpation behavior of the participants is adapted to localize the nodule or to estimate the depth. From these observations and results, we found interesting to compare two types of sweeping strategies, one local with a light force applied to the phantom using the tip of the probe and one more global using the whole palmar region of the probe with the tactile sensor.

The first sweeping direction is the lateral sweep (along **y**). The aim of this sweeping strategy to perform the nodule detection on a wide surface. Therefore, during lateral sweeps, the probe is first significantly indented (along **z**) in the phantom to have contact between the whole palmar surface of the VLM probe and the phantom (see Fig 2).

The second sweeping strategy, denoted longitudinal sweep, aims to improve the depth estimation of a hard nodule by performing a localized exploration. Indeed, during this longitudinal sweeps, the probe only slightly touches the surface of the phantom with the edge of its tip and is swept along the **x** direction.

The trajectories of the VLM probe during the lateral sweep and longitudinal analysis are shown in Fig 2.

During the study of the lateral sweeps, the probe is initially positioned to align the nodule with the end of the probe's tip and to have the nodule roughly centered on the trajectory. The probe is first indented at 25mm, then it is swept along **y** axis by 120mm at 30mm/s. The probe is then outdented and moved back at 100mm/s to the original position. This cycle is repeated 5 times, and after the fifth time, the VLM probe is shifted by 5mm along **x** axis to a new initial

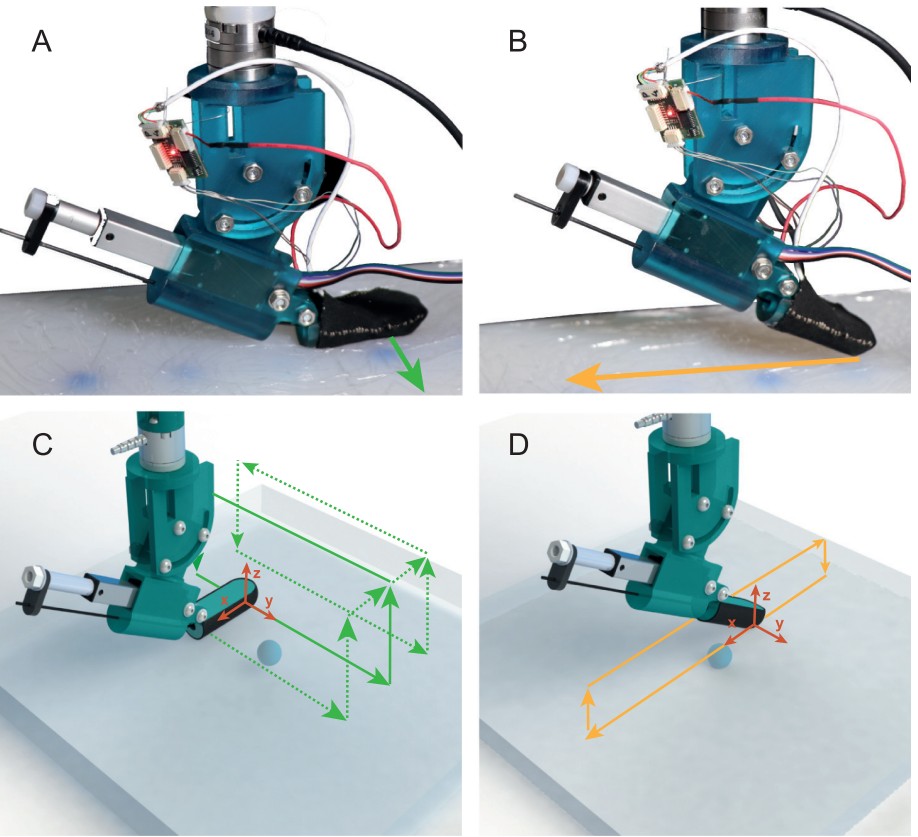

**Fig 2. Sweeping directions.** (A) Picture taken during a lateral sweep. (B) Picture taken during a longitudinal sweep. (C) Probe trajectory during lateral sweeps with different shifts. (D) Probe trajectory during longitudinal sweeps.

position. As the lateral sweeps are performed to localize nodule on a wide area, the aim of this shift is to observe the behavior of the probe when the latter is sweeping over a nodule at different distances. The next cycle is also repeated 5 times before applying a new shift. In total, 4 shifts are applied, the distance between the initial and last trajectories is then 20$mm$.

During the study of the longitudinal sweeps, the probe is first indented by 4mm. With a 4mm indentation, only the tip's edge of the probe is touching the phantom. The probe is then swept along the **x** axis by 120mm. The speed of the sweep is also set at 30mm/s. The probe has been initially positioned to sweep over the center of the nodules placed in the silicone. Finally, the probe is lifted up to avoid contact between the phantom and the probe and moved back to the initial position. The longitudinal sweeping is repeated 25 times.

The two sweeping directions have been tested for 15 different stiffnesses covering the range of stiffness that the probe can achieved with a carbon rod of 1.5mm.: 0.65, 0.66, 0.67, 0.68, 0.69, 0.7, 0.71, 0.73, 0.74, 0.76, 0.77, 0.8, 0.83, 0.87, and 0.94Nm/rad and for 4 different nodule depths (2, 4, 6, and 8mm) and without nodule for a total of 1875 trials per sweeping direction. The supplemental S1 and S2 Videos show respectively some lateral sweeps and some longitudinal sweeps.

One can notice that the steps of the stiffness tested in this paper is not linear. This comes from the fact that for simplicity, we have chosen linear steps of 2mm in the active length of the carbon rod. This choice allows us to take advantage of the probe characterization performed in our previous study [22] and makes the stiffness control easier by relying on the closed-loop position control of the linear actuator. It also simplifies the implementation of carbon rod

displacement in the FE simulation. However, since the relation between the stiffness and the active length of the carbon rod is nonlinear, it results in nonlinear steps of stiffness.

## Finite Element simulation

The aim of our Finite Element (FE) model is to provide a further study on the impact of the joint stiffness variation during palpation exploration. The experimental results show that the variation of stiffness is more significant for the longitudinal sweeps than for the lateral sweep. As a consequence, we focused our FE simulation on longitudinal sweeps.

Using *COMSOL multiphysics*, a 2D Finite Element (FE) model has been developed to simulate the VLM probe and phantom behavior during the longitudinal sweeps. To reduce the complexity of this simulation performed the geometry of the probe has been simplified. Fig 3 presents the mesh and the different material domains of the FE simulation.

All the materials except the phantom silicone have been modeled as linear elastic materials. The *Ecoflex 0010* material, as several rubber silicones, follows a nonlinear behavior when it is significantly deformed. This is the reason why the material of the phantom has been modeled with a hyperelastic model in addition to a viscoelastic model. The chosen hyperelastic model is the Ogden model with the parameters obtained by Spark *et al.* [50]. The viscoelastic model is based on a Standard Linear Solid (SLS) model. The parameters have been obtained from experimental characterization during a previous study [11].

The bottom part of the phantom is constrained in position whereas a prescribed displacement is applied to the probe. To save some computing time, the sweeping distance has been reduced to 60mm (120mm in the experiments). The simulation has been performed for 15 active lengths of carbon rod (from 24mm to 52mm in steps of 2mm) which are equivalent to the 15 levels of stiffness studied in the experiment.

The contact between the probe and the phantom is modeled with two surface contact pairs covering the palmar region of the probe and the upper layer of the phantom, respectively. The pressure contact calculation is based on an Augmented Lagrangian Method, and the friction between the 2 contacts is modeled as Coulomb friction ($\mu$ in Table 1). Finally, no rolling resistance is modeled for the contact between the probe and the phantom assuming pure sliding at the elements level.

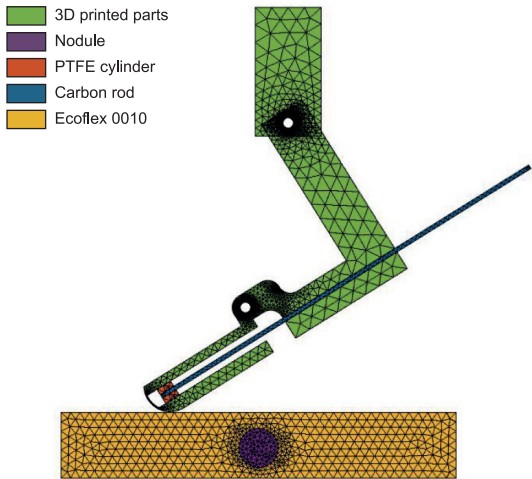

**Fig 3. FE simulation mesh and materials.** 6mm deep nodule and an active length of carbon rod of 52mm (equivalent to a stiffness of 0.65Nm/rad).

**Table 1. Simulation parameters.**

| Domain | Parameter | Description | Value | Unit |
|---|---|---|---|---|
| General | $v_x$ | Sweeping velocity | 30 | mm/s |
| | $t_{indent}$ | Time to reach the maximum indentation | 0.6 | s |
| | $\Delta_t$ | Time step of the simulation | $10^{-2}$ | s |
| | $z_{max}$ | Indentation | 4 | mm |
| | $\mu$ | Static Coulomb friction coefficient between the phantom and the probe | 0.1 | SI |
| 3D printed parts | $E_1$ | Young modulus | $2.7 \times 10^9$ | Pa |
| | $\rho_1$ | Density | 1250 | kg/m$^3$ |
| | $v_1$ | Poisson's ratio | 0.36 | SI |
| Nodule | $E_2$ | Young modulus | $3.2 \times 10^9$ | Pa |
| | $\rho_2$ | Density | 1180 | kg/m$^3$ |
| | $v_2$ | Poisson's ratio | 0.37 | SI |
| PTFE cylinder | $E_3$ | Young modulus | $4 \times 10^8$ | Pa |
| | $\rho_3$ | Density | 2200 | kg/m$^3$ |
| | $v_3$ | Poisson's ratio | 0.48 | SI |
| Carbon rod | $E_4$ | Young modulus | $10.2 \times 10^9$ | Pa |
| | $\rho_4$ | Density | 1000 | kg/m$^3$ |
| | $v_4$ | Poisson's ratio | 0.49 | SI |
| Ecoflex 0010 | $G_5$ | Shear modulus (Ogden model) | 12605 | Pa |
| | $\alpha_5$ | Strain Hardening Exponent (Ogden model) | 4.32 | SI |
| | $\tau_5$ | Relaxation time (SLS viscoelastic model) | 2.30 | s |
| | $\beta_5$ | Energy factor (SLS viscoelastic model) | 0.6 | SI |

The mesh of the FE model is composed of triangular elements (3933 elements for simulations with nodules and 3566 for the simulation without nodule). To increase the accuracy, a mesh refinement has been done around the nodule and around the probe joints. About 27 hours of computations on a 2.6GHz *Intel Xeon* 16 cores machine with 128GB of RAM were needed to simulate all the configurations proposed in the paper.

The main simulation parameters are given in Table 1.

## Results

In this section, the results obtained during the experiments and from the FE simulation are presented. These results illustrate the differences between the two sweeping directions as well as the role of the VLM probe's stiffness variation on haptic perception. In the last subsection, we also describe how the likelihood functions of the force knowing the depth are generated. Thus, these likelihood functions can be seen as a knowledge obtain from past palpation or from the simulation and are used in the algorithm proposed in this paper.

### Lateral sweep

The aim of this experiment is to analyze the suitability of the VLM probe for detecting a nodule concealed in soft tissues during a lateral sweep. The objective is also to understand if the stiffness is playing a role in nodule detection. Finally, it is explained how the bi-modal sensing (kinesthetic and tactile) helps in the localization of the nodule.

Fig 4 shows an example of the signals acquired during lateral sweep experiments on a phantom without any nodule and with 16mm diameter nodules placed respectively at 2mm, 4mm, 6mm and 8mm from the surface. The signals shown on this figure are the force measured by

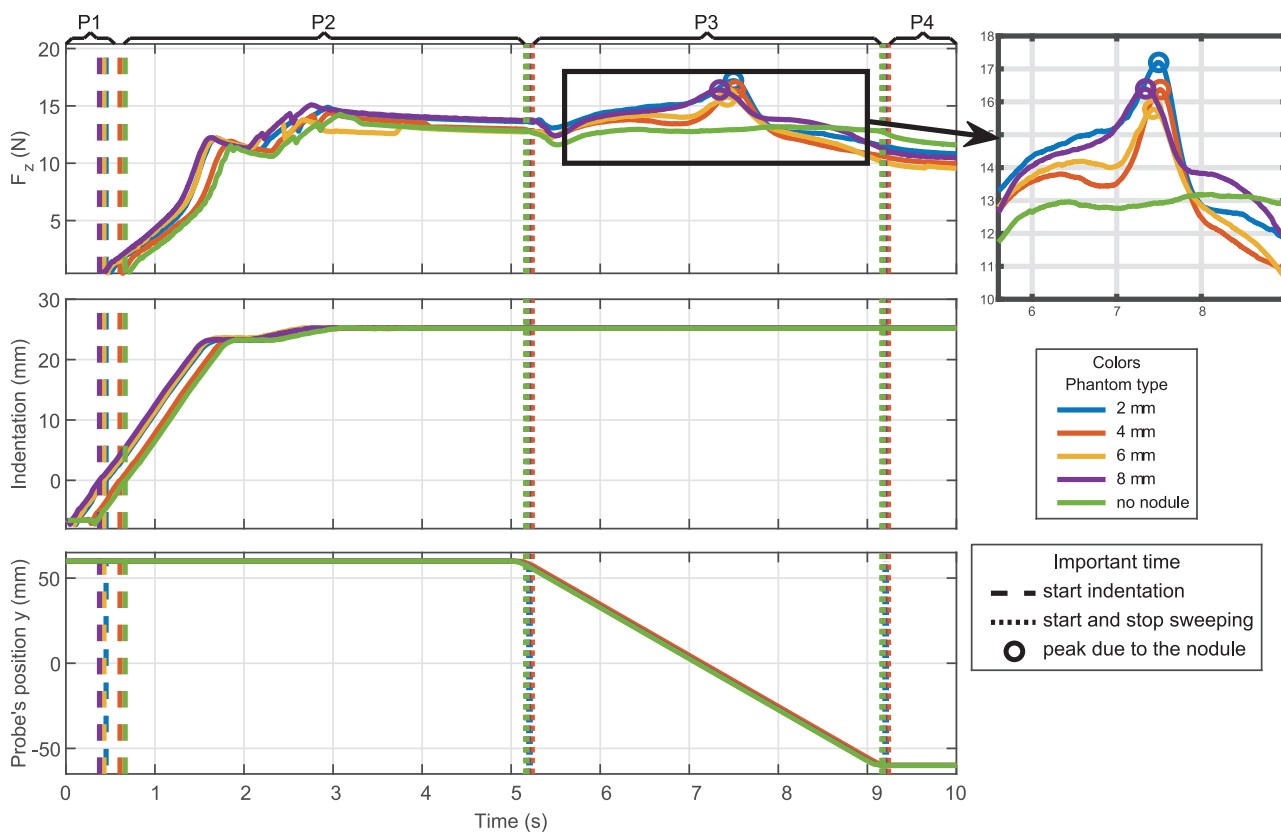

**Fig 4. Data acquired during one lateral sweep.** Stiffness = 0.65Nm/rad and 20mm shift.

the force/torque sensor along the **z** axis filtered with a Savitzky-Golay filter, the indentation and the $y$ position of the probe. In this figure, four main periods can be distinguished:

**P1**: From $t$ = 0s to $t \approx$ 0.6s. During this period, the VLM probe moves along the **z** axis but the probe is not in contact with the phantom, the indentation is then considered negative.

**P2**: From $t \approx$ 0.6s to $t \approx$ 5.2s. This is the indentation period. The probe continues moving along the **z** axis to reach an indentation of 25 mm. One can notice that the speed of the indentation reduces after 20mm of indentation, this phenomenon is due to the reaction force of the phantom and the frictions of the linear actuator and mechanism which counteract the proportional component of the regulator. It is then the integral action of the integrated controller which helps to converge to the desired position, this integral gain is set at a high value to ensure the convergence to the desired position.

**P3**: From $t \approx$ 5.2s and $t \approx$ 9.2s. This is the sweeping period. The probe moves along the **y** axis by 120mm with a speed of 30mm/s. It is important to take into account that since the acceleration and deceleration period are small, they are neglected in the rest of the paper. It can be noticed that at the beginning of the sweeping period, the force along **z** axis drops suddenly. This force drop can be explained by the action of several phenomena, in particular, the dynamic forces due to the acceleration but also the compliance of the joints and links. Indeed, all the joints except the variable stiffness joint are assumed to be rigid, but due to the play between some parts, the probe rotates slightly around the **y** axis which releases a bit the normal force applied to the phantom.

**P4**: The last period is for $t \geq 9.2$s. The probe is static still indented. One can notice a small relaxation period after the probe stops.

Fig 4 shows that a force peak on **z** axis is measured when the probe sweeps over a nodule. This peak can easily be detected since it is also the maximum value measured during the period P3. It can be noticed that the force peak value is not necessarily suitable to estimate the depth of the nodule. Thus, in Fig 4, the force peak value for the nodule 8mm deep is higher than the force peak value for the nodule 4mm deep. This can be explained by the fact that a small error on the indentation or a small variation on the thickness of the phantom implies a big variation of the value. After studying the results of several trials, it seems more reliable to study the prominence of the peak. In this paper, we define the prominence of the peak as the difference between the force peak value and the force baseline value at the corresponding time of the peak. The baseline computation method used in this study is based on the Asymmetric Least Squares (ALS) method which is suitable for detecting a baseline of a signal with peaks [51]. The ALS baseline parameters have been tuned for the VLM probe signals but are kept constant for all the experiments presented in this paper.

Fig 5 illustrates the force measured, the baseline computed, and the prominence of the peak for several combinations of stiffnesses and shifts when the probe is sweeping laterally over the

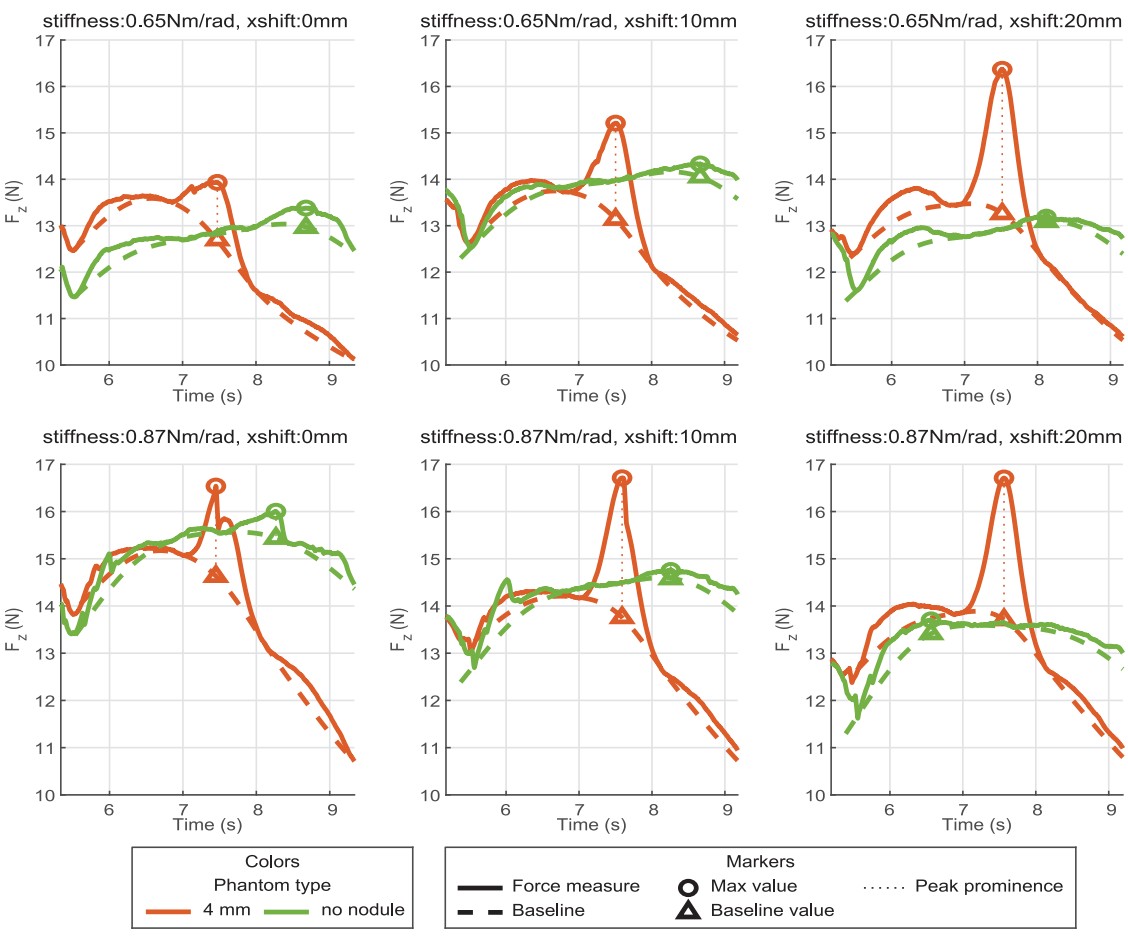

**Fig 5. Force, baseline, and force peak prominence acquired for several shifts.**

nodule concealed at 4mm and the part of the phantom without nodule respectively. It has to be noticed that even if there is no peak due to a nodule, a peak prominence is computed from the sweep over the phantom without nodule. As for the other phantom sample, the prominence is computed from the distance between the baseline and the maximum force value detected. The peak prominences measured for the nodule are clearly higher than the prominences for the lateral sweep where no nodule is concealed.

By comparing the different rows of Fig 5, one can see that the maximum force and the prominence increase with the shift. This can be explained by the fact that in the range covered in this paper, the higher the shift is, the closer the nodule is to the VLM probe joint. Then for the same tangent displacement (along **z**), if the point of application is closer to the center of rotation, the equivalent angular displacement is higher. Finally, if the angular displacement is higher, due to the stiffness, the reaction torque and forces applied are greater.

Fig 6 shows the distributions of the force peak prominence per stiffness for each depth of nodule. It has to be noticed that the graph does not separate the shift so each box is obtained from 25 trials. These distributions presented with a boxplot standard confirm that there is a clear difference for the sweep with and without nodule. This figure also shows that the boxes for the different nodule depths are overlapping, this implies that from a single measure, it would be difficult to determine the nodule depth. To further support the interpretation of Fig 6, we detail, in the supplemental S1 Appendix, a comparison of the distributions obtained for each stiffness using statistical analysis.

However, Fig 7A shows the contact pressure measured by several taxels of the Cyskin sensor during lateral sweeps over a 2mm deep nodule for different shifts. It can be seen on this figure that depending on the shift, a peak of contact pressure is measured when the probe sweeps over the nodule. Fig 7B exhibits the pressure contact prominence map measured by the tactile sensor at the force peak instant for different shifts. The latter shows once again that the region

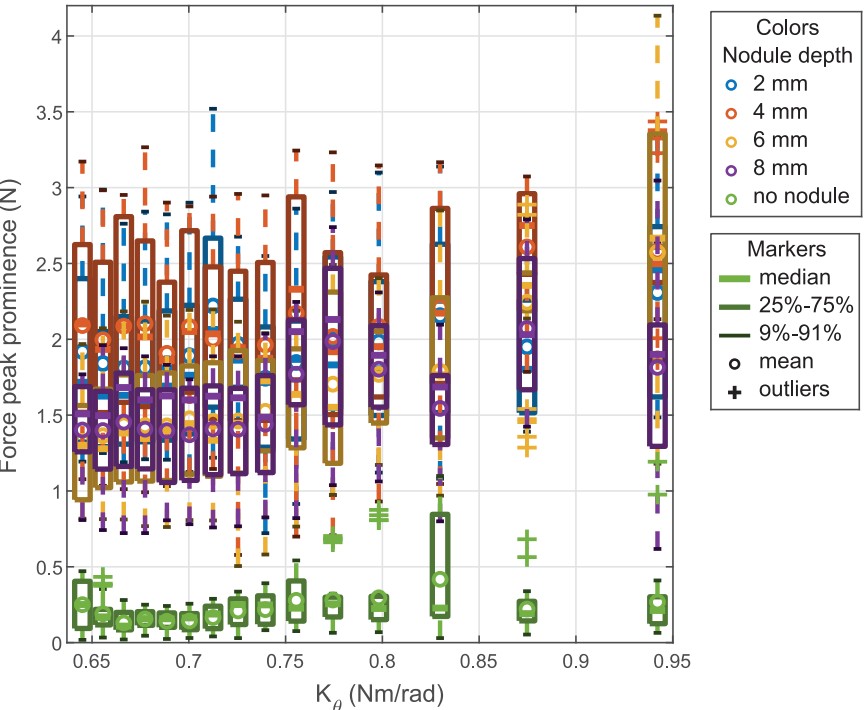

**Fig 6. Force peak prominence per stiffness for lateral sweeps.**

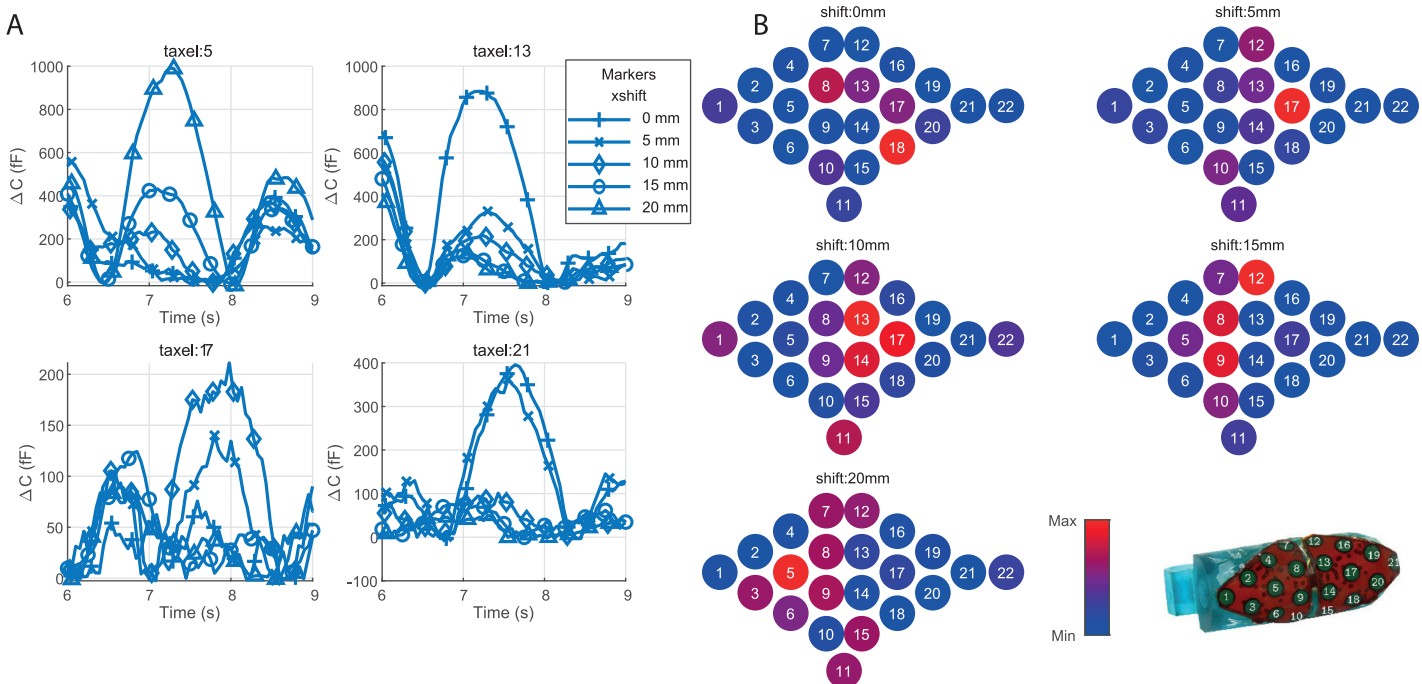

**Fig 7. Tactile sensor data.** (A) Tactile data after baseline correction per shift. (B) Normalized tactile prominence map computed at the peak time per shift.

where the max contact pressure prominence is measured depends on the shift. By knowing the probe position and geometry, the localization of the nodule on **x** axis can then be estimated by interpolation. The position of the nodule on **y** axis is directly obtained from the position of the probe at the force peak instant.

## Longitudinal sweep

The previous section shows that the lateral sweep explores a large area and helps to find the location of the nodule but it only gives a rough estimate of the nodule depth. In this section, the simulation and experimental results for the longitudinal sweep that allows exploring the phantom more locally are presented. These results show that the longitudinal sweep is more suitable for estimating the depth of the nodule and highlights the role of the stiffness on haptic perception.

**Simulation results.** Due to the nonlinear behavior of the soft material, coming from its hyperelasticity and its viscoelasticity, it is not simple to obtain an analytical model for the phantom. FE simulation is a way to have a better insight into what happens at the tissue level.

The supplementary S1 Fig and S3 Video shows some simulation frames for different nodule depth and stiffnesses. The stress in the soft tissues increases when the probe sweeps over the nodule. As can be seen, the deeper the nodule is, the smaller the stress is. Moreover, with the small forces applied during the longitudinal sweeps, the stress in the material under the nodule is not impacted as much as the stress in the material above the nodule. This phenomenon comes from the fact that during the palpation sweeps, the displacement of the nodule is small compared to the displacement of tissue above the nodule. This shows that the probe is more significantly affected by the amount of material above the nodule (the nodule depth) than the amount of material under the nodule.

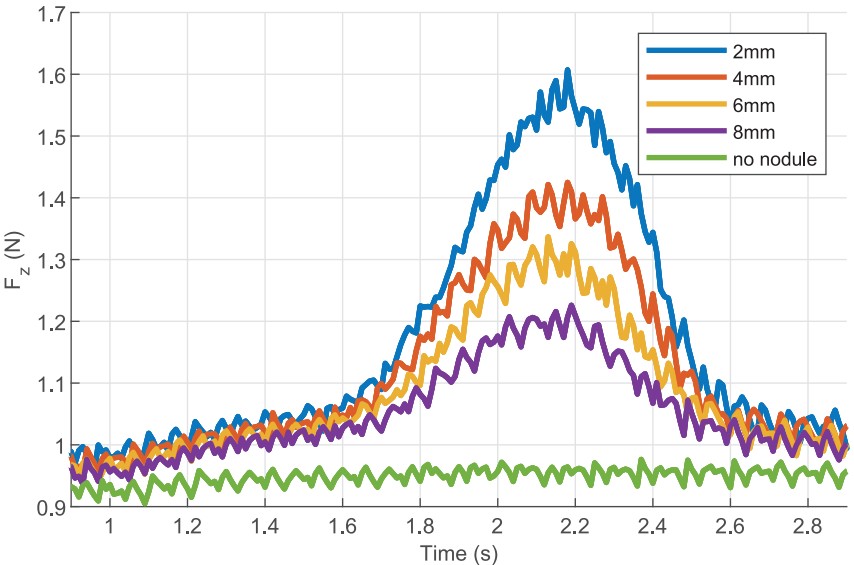

**Fig 8. Force simulated for longitudinal sweeps with a stiffness of 0.66Nm/rad.**

From the FE model, the kinesthetic force measured by the F/T sensor can also be simulated. Fig 8 shows an example of the evolution of the force during the sweeps for the same stiffness. It can be seen that due to the dynamics of the contact and the nonlinear behavior of the tissue, some oscillations appear in the simulated force signals. These oscillations are due to a stick-and-slip behavior coming from the friction between the probe and the phantom. These oscillations are more or less filtered by the mechanical impedance of the probe in which the stiffness plays an essential role. These oscillations can be observed on force signals measured during the experiments (see Fig 10). Furthermore, one can notice that the amplitude of these oscillations varies with the nodule depth, which means that these oscillations are not only dependent on the probe's internal dynamics but also on the ones from the phantom.

Fig 9 shows the distributions of the force peak prominence per stiffness for the different nodule depth obtained from the FE simulations. One can notice that this time the distributions for the different nodule depth can be distinguished from one to another. The simulation results are compared to the experimental results in the next subsection.

**Experimental results.**   The force data presented in Fig 10 shows once again that a peak can be detected when the probe sweeps over the nodule. This figure also shows that the force peak prominence seems more reliable than simply the peak height. For the longitudinal sweep, the peak detection is also simple since it corresponds to a maximum force measured during each trial.

Fig 11 presents with boxplot format the distribution of the force peak prominence for the 15 different stiffnesses and each nodule depth. It can be seen that compared to the results obtained for the lateral sweep (Fig 6) the distributions can be distinguished the majority of the cases. This observation confirms the results obtained from the FE simulations. Similarly to the lateral sweeps, the significance of the probe's stiffness variation on the force peak prominence distribution is further studied, using statistical analysis, in the supplemental S1 Appendix.

The stiffness plays a role in the width of the distributions but also on the distance between the distributions. To compare the distance between the distributions for a given stiffness, the

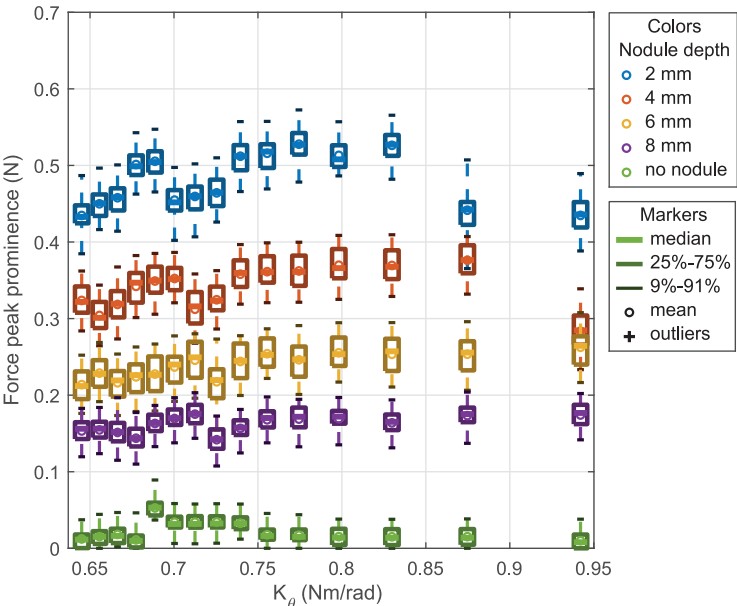

**Fig 9. Peak amplitude vs stiffness obtained in simulation.**

Standardized Euclidean Distance (SED) is used. The SED can be defined as follows:

$$d_{\mathrm{SED}}(x, y) = \sqrt{\frac{(x-y)^{\mathsf{T}}(x-y)}{\sigma_x \sigma_y}}, \tag{1}$$

where $x$ and $y$ denotes two random vectors describing the distributions to be compared. $\sigma_x$ and $\sigma_y$ are the standard deviation of the random vectors $x$ and $y$ respectively. This metric is

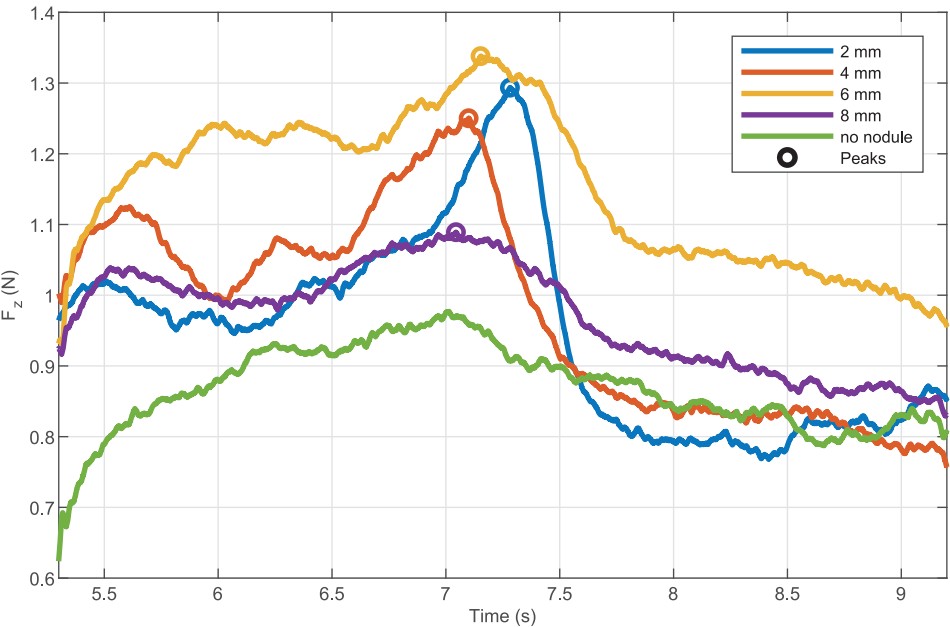

**Fig 10. Data acquired during longitudinal sweeps for the different nodule depth.**

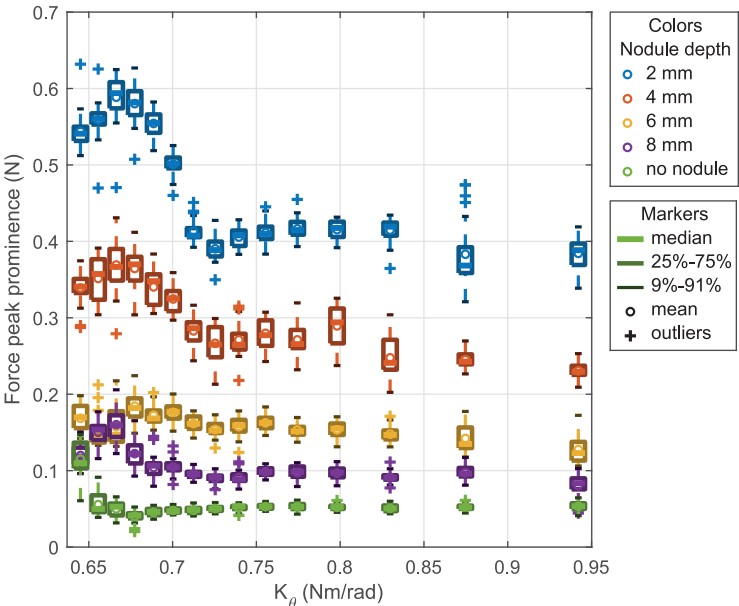

**Fig 11. Force peak prominence vs stiffness for longitudinal sweeps.**

suitable to compare two distributions taking into account the standard deviation [52]. One can notice that the two vectors *x* and *y* must have the same dimension.

Fig 12A gives the minimum SED between the force peak prominence distribution without nodule and the force peak prominence distribution with nodule across the different stiffnesses of the probe. The results show that the distance varies depending on the stiffness. The higher this SED is the easiest it is to distinguish if there is a nodule. The figure shows that both sweeping directions give good results to distinguish if there is a nodule or not. In particular, the bests

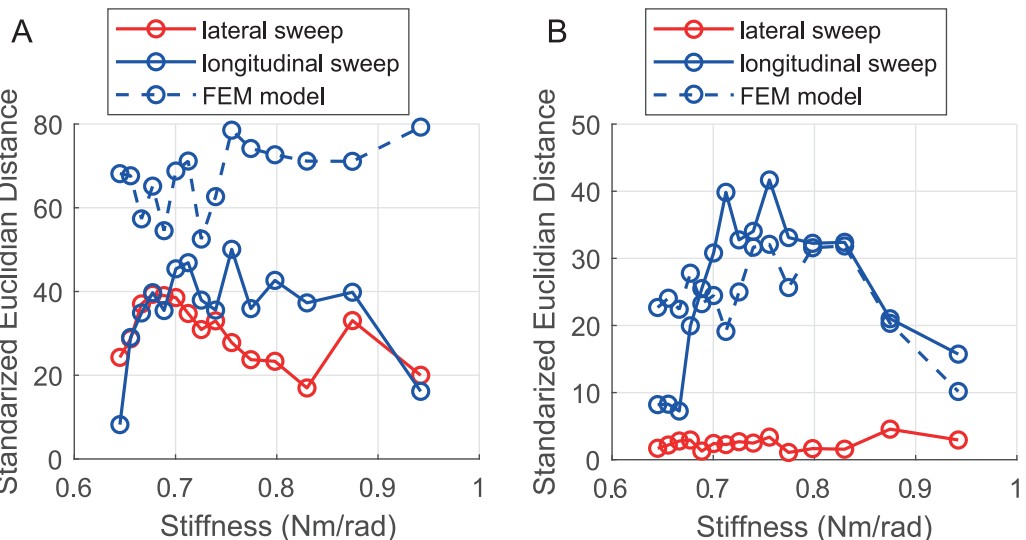

**Fig 12. SED evaluations.** (A) Minimal distance between the distributions with and without nodule. (B) Minimal distance between the distributions with nodule.

of the tested stiffnesses to detect the presence of a nodule in the stiffness range of the VLM probe are respectively $K_\theta$ = 0.68Nm/rad and $K_\theta$ = 0.76Nm/rad for the lateral sweep and for the longitudinal sweep. It can be noticed that since the longitudinal sweep is inspecting the phantom more locally, the distance between the force peak prominence distribution with nodule and without nodule is slightly higher, in particular for medium stiffness. Finally, the SED distance obtained from the FE simulation is overestimated. This can be explained by the fact that in the experimental approach, the phantom is not perfectly flat in contrast to the FE simulation where the phantom is virtually perfectly flat. This difference can also be observed by comparing the force peak prominence distributions, where it can be seen that the mean force peak prominence obtained during the simulation is closer to zero than the one obtained experimentally.

Fig 12B presents the minimum SED distance between two distributions of the Force peak prominence when a nodule is present. This time, the minimum distance measures the ability to distinguish the depth of a nodule for the different stiffnesses. Thus, a clear difference between the two sweeping directions is observed. Indeed, Lateral sweep minimum is smaller than the longitudinal one. This highlights that a local investigation simplifies the nodule depth distinction. Finally, the SED distance obtained from the FE simulations gives a good insight into the role of the stiffnesses in the nodule depth distinction.

### Likelihood functions

From the previous experimental and simulation results, we can compute for each stiffness the probability density functions of the force peak prominence knowing the depth of the nodule: $p^{lat}_{K_\theta}(F|d)$ and $p^{long}_{K_\theta}(F|d)$ for the lateral sweep and the longitudinal sweep respectively. Where $F$ refers to the force peak prominence, $d$ refers to the depth of the nodule and $K_\theta$ refers to the stiffness of the probe. These functions are the results of a knowledge obtained by past or simulated explorations of the phantom. They are used in the nodule detection algorithm to update the depth estimation with Bayes inference.

To obtain the likelihood functions, we used a non-parametric Kernel distribution fitting using *Matlab 2018*. Fig 13A and 13C illustrate the obtained likelihood functions for each stiffness. The role of the stiffness on the sharpness of the distribution, as mentioned earlier, can also be seen in these figures.

These likelihood functions are computed for the known nodule depth $d \in \{2, 4, 6, 8\}$ and without nodule. Using these functions would be sufficient to estimate the depth of the nodule with a 2mm accuracy. However, to increase the resolution of the estimation, the probability density functions have been interpolated by linear interpolation of uncertain data [53]. The interpolated functions are shown in Fig 13B and 13D for the experimental results and on Fig 13E for the one generated from the simulation results.

### Nodule detection algorithm

In this section, we propose the algorithm 1 for nodule detection that uses the property of the stiffness to condition the likelihood of the force peak prominence. This algorithm, based on Bayes inference and information gain theory, controls the VLM probe and its stiffness to autonomously palpate soft tissues and determine if there is or not a nodule. Moreover, the algorithm returns the position of the nodule in the (**x**,**y**) plane and the estimated depth. All the notations used in this algorithm and section are described in Table 2.

From the results described in the previous section, we obtained the probability density functions of the force knowing the depth for both sweeping directions and for each stiffness denoted $p^{lat}_{K_\theta}(F|d)$ and $p^{long}_{K_\theta}(F|d)$ respectively.

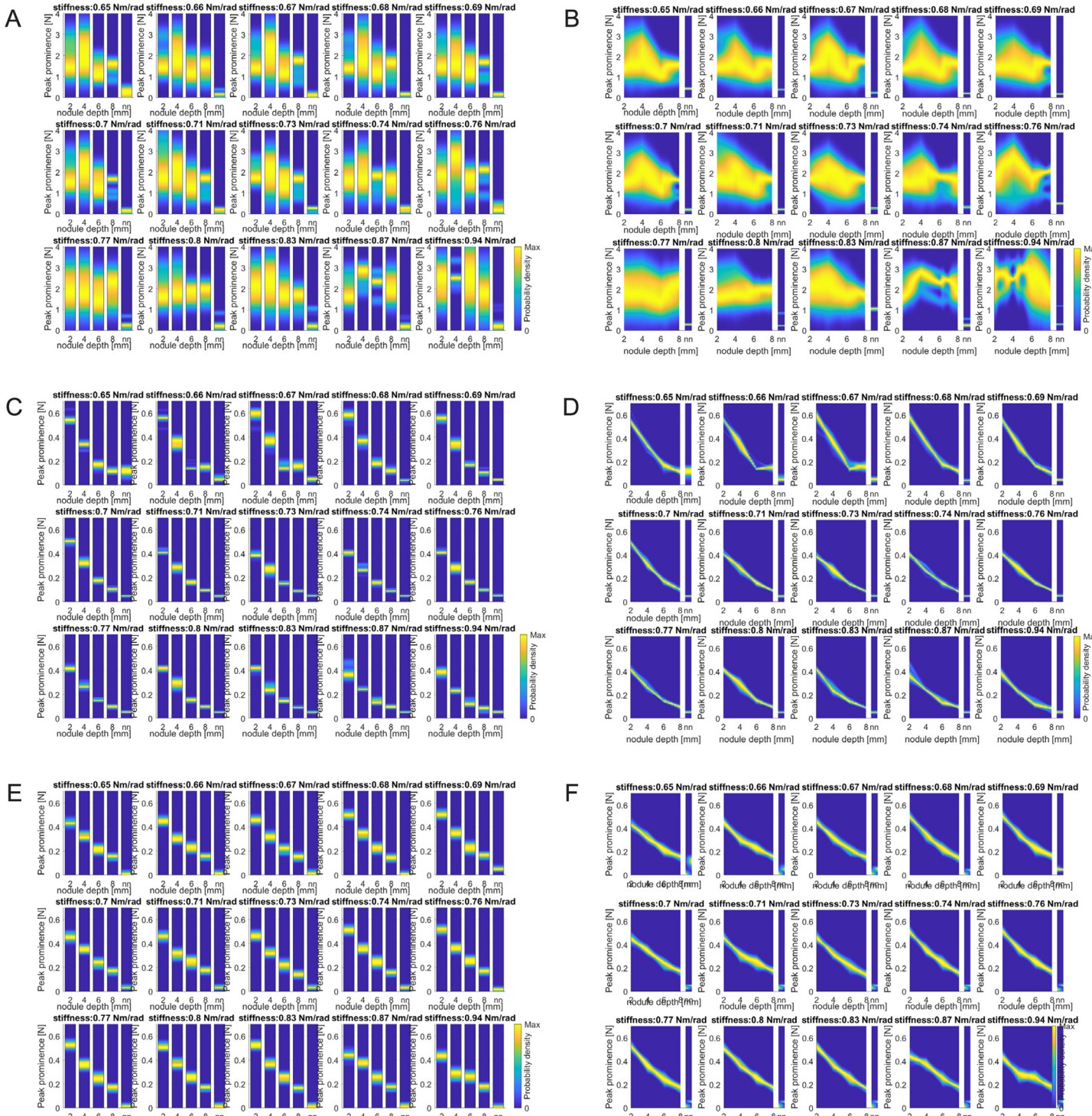

**Fig 13. Likelihood functions.** (A) Probability density functions for lateral sweep. (B) Interpolated probability density functions for lateral sweep. (C) Probability density functions for longitudinal sweep. (D) Interpolated probability density functions for longitudinal sweep. (E) Probability density functions for longitudinal sweep from FE simulation. (F) Interpolated probability density functions for longitudinal sweep from FE simulation. nn refers to no nodule. In order to highlight the variation of sharpness of the probability density functions the yellow color is attributed to the maximal value of each vertical strip.

**Table 2. Algorithm parameters.**

| Notations | Description |
|---|---|
| $p^{lat}_{K_\theta}(F\|d)$ | Probability density function of the force peak knowing the depth of the nodule $d$ for a lateral sweep at stiffness $K_\theta$ |
| $p^{long}_{K_\theta}(F\|d)$ | Probability density function of the force peak knowing the depth of the nodule $d$ for a lateral sweep at stiffness $K_\theta$ |
| $P(N)$ | Probability of the presence of a nodule in the explored area. |
| $P(d)$ | Probability mass function of the depth of the nodule in the explored region. |
| $P(N\|F)$ | Posterior estimation of the probability of the presence of the nodule knowing the force peak prominence of the last sweep |
| $P(d\|F)$ | Posterior estimation of the probability mass function of the depth of the nodule knowing the force peak prominence of the last sweep |
| $dir \in \{lat, long\}$ | Direction of sweeping; lateral or longitudinal |
| $p^{dir}_{K_\theta}(F_{K_\theta})$ | Estimated probability density function of the force for the next sweep with the stiffness $K_\theta$. |
| $P^{th}_N$ | Threshold on the probability of the presence of a nodule |
| $P^{th}_d$ | Threshold on the probability of depth of a nodule |
| $x_n$ | Position of the nodule on the **x** axis. |
| $y_n$ | Position of the nodule on the **y** axis. |
| $IG$ | Information gain of the sweep, computed from the Kullback-Leibler divergence |
| $IG^{th}$ | Threshold on the information gain. |
| $K_\theta$ | Stiffness of the probe for the sweep |
| $K^{detect}_\theta$ | Best stiffness of the probe for distinguishing if there is a nodule |
| $\mathcal{K}$ | Set of stiffness values $K_\theta$ that the probe can take. For the VLM probe $\mathcal{K} = \{0.65, 0.66, 0.67, 0.68, 0.69, 0.7, 0.71, 0.73, 0.74, 0.76, 0.77, 0.8, 0.83, 0.87, and 0.94\}$ |
| $\mathcal{D}$ | Set of depth considered for the depth of the nodule. In this study $\mathcal{D} = [2 : 0.2 : 8]$ |

## Description of the algorithm

### Algorithm 1: Proposed algorithm for nodule detection

```
Data: Likelihood functions P^lat_Kθ(F|d) and P^long_Kθ(F|d)
Result: Probability of the presence of a nodule P(N), Probability mass
function of the depth of the nodule P(d), and the coordinates of the
position of the nodule (xn, yn)
1  Initialization
2  do
3    if P(N) < P^th_N then
4      the probe sweeps laterally over the region to explore with the
       stiffness Kθ = K^detect_θ
5    else
6      foreach Kθ ∈ K do
7        Compute p_Kθ(F_Kθ)
8      end
9      the probe sweeps longitudinally over the region of the nodule
       with the stiffness Kθ = arg min_{Kθ∈K}(Var(F_Kθ))
10     end
11   Filter force and tactile data
12   Find maximum force during the sweep;
13   Compute prominence;
14   if P(N) < P^th_N then
15     Compute posterior estimations P(d|F) and P(N|F) with p^lat_Kθ(F|d)
```

```
16    Compute information gain IG = D_KL(P(N|F) ‖ P(N))
17  else
18    Compute posterior estimations P(d|F) and P(N|F) with p_Kθ^long(F|d).
      Compute information gain IG = D_KL(P(d|F) ‖ P(d))
19  end
20  if P(N|F) < P_N^th then
21    Update nodule position (x_n,y_n) from probe position and tactile
      measurement at maximum force
22  end
23  Update P(N) and P(d) with the posteriori estimations
24 while P(N) > 1 − P_N^th & max_{d∈D}(P(d)) < P_d^th & IG > IG^th;
25 return P(N), P(d), x_n and y_n
```

Line 1 of the algorithm 1 refers to the initialization. During this step, the probe is manually positioned over the region of soft tissue that we want to explore. The probability of the presence of the nodule $P(N)$ is set at 0.5 (same probability between the presence of a nodule and the non-presence of a nodule). The probability mass function of the nodule depth is set as flat distribution.

The condition in line 3 tests the likelihood of the presence of the nodule. If the probability $P(N)$ is high enough, it means that a nodule has been detected during the precedent sweeps and that its position has been estimated. A longitudinal sweep is then performed over the region of the nodule to give a better estimation of the depth. On the other hand, if $P(N)$ is still lower than the threshold, a lateral sweep is performed over the region to explore.

The stiffness chosen for the sweep is dependent on the direction and the depth estimation $P(d)$. For lateral sweeps, the stiffness chosen is $K_\theta^{detect} = 0.68$ Nm/rad which is the stiffness that maximizes the SED between the distributions with nodule and the distribution without nodule during lateral sweep as shown on Fig 12A. For longitudinal sweeps the stiffness chosen is the one which minimizes the variance of the estimated peak force prominences $F_{K_\theta}$ with the probability distributions computed for all $K_\theta$ in $\mathcal{K}$ as follows:

$$p_{K_\theta}^{dir}(F_{K_\theta}) = \sum_{d\in\mathcal{D}} P(d)p_{K_\theta}^{dir}(F|d) \tag{2}$$

The filters used at the line 11 are the same Savitzky-Golay filters used for post-processing the data in the results section. Similarly, the prominence is computed by subtracting the ALS baseline to the maximum force measured during the sweep.

Depending on the direction of the sweep the posterior estimation of the presence of a nodule and the probability mass function of the depth are computed with the Bayes inferences as follows:

$$
\begin{aligned}
P(N|F) &= \frac{P(N)p_{K_\theta}^{dir}(F|N)}{P(N)p_{K_\theta}^{dir}(F|N) + (1 - P(N))p_{K_\theta}^{dir}(F_{K_\theta})} \\[2mm]
P(d|F) &= \frac{P(d)p_{K_\theta}^{dir}(F|d)}{p_{K_\theta}^{dir}(F_{K_\theta})}
\end{aligned}
\tag{3}
$$

where the probability density functions are evaluated for the force peak prominence measured during the sweep and $dir \in \{long, lat\}$ refers to the direction.

It can be noticed that $p_{K_\theta}^{dir}(F_{K_\theta})$ appears in the denominator of the Bayes inference in (3). Minimizing the variance of the expected force peak prominence at the line 9 increases the probability of having a sharp posterior estimation of the depth.

The information gain denoted *IG* is computed depending on the direction of the sweep using the Kullback-Leibler (KL) divergence which is a metric to evaluate how much information is gained when the probabilities $P(N)$ and $P(d)$ are updated from the prior to the posteriors. The KL divergence is often used in machine learning or with Bayes inference to evaluate the gain of information obtained with the update of the probability distributions. The KL divergences used in line 16 and 18 are given by the following equation:

$$D_{KL}(P(N|F)\|P(N)) = P(N|F)\log\left(\frac{P(N|F)}{P(N)}\right)$$

$$+(1-P(N|F))\log\left(\frac{1-P(N|F)}{1-P(N)}\right) \quad (4)$$

$$D_{KL}(P(d|F)\|P(d)) = \sum_{d\in\mathcal{D}}P(d|F)\log\left(\frac{P(d|F)}{P(d)}\right)$$

If the posterior estimation of the presence of the nodule is higher than the threshold, the position of the nodule in the $(\mathbf{x}, \mathbf{y})$ plane is updated. If the sweep is a lateral sweep, the position $y_n$ is updated with the position where the maximum kinesthetic force has been detected, whereas $x_n$ is estimated from the tactile sensor measures. Thus, the position is estimated from the normalized tactile prominence map at the time of the peak (as shown in Fig 7B). If the sweep is a longitudinal sweep, only the position $x_n$ is updated from the position where the maximum force has been detected.

Line 24 gives the conditions which determine if the algorithm should continue sweeping. In other words, another sweep is needed if: 1) it is likely to have a nodule 2) the maximum probability of the depth $P(d)$ is too low 3) the information gain of the last sweep is high enough. The respective thresholds $P_N^{th}$, $P_d^{th}$, and $IG^{th}$ can be tuned to change the performances of the algorithm. Indeed, augmenting $P_N^{th}$, for instance, increases the confidence on the nodule detection but is likely to increase the number of lateral sweeps needed. $P_d^{th}$ and $IG^{th}$ allows tuning the confidence on the depth estimation of the nodule and the minimal information gain respectively.

Several values of thresholds have been tried experimentally. $P_N^{th} = 0.8$, $P_d^{th} = 0.7$, and $IG^{th} = 0.02$ have been found to provide a good trade-off between the confidence level and number of sweeps. These parameters' values are the ones that have been used for the evaluation of the algorithm.

## Evaluation of the algorithm

The evaluation of the proposed algorithm was carried out using the VLM probe. The evaluation was first performed using the likelihood function obtained experimentally, and then with the likelihood function generated from the FE simulations. It has to be noticed that for the 2mm and 4mm depth, the tests were done using the nodules which have not been used for computing the likelihood functions. Finally, the proposed palpation strategy aims to localize the nodule independently from the phantom orientation, so the algorithm has been tested for several orientations of the phantom.

Fig 14A shows representative results from the several trials performed with the algorithm. This figure shows the evolution of the probability mass function of the depth $P(d)$ after each sweep. On average on all trials, the final expected value of the depth is in a range of 0.5mm around the real depth. The median of the number of sweeps is 5, with a minimum of 3 and a

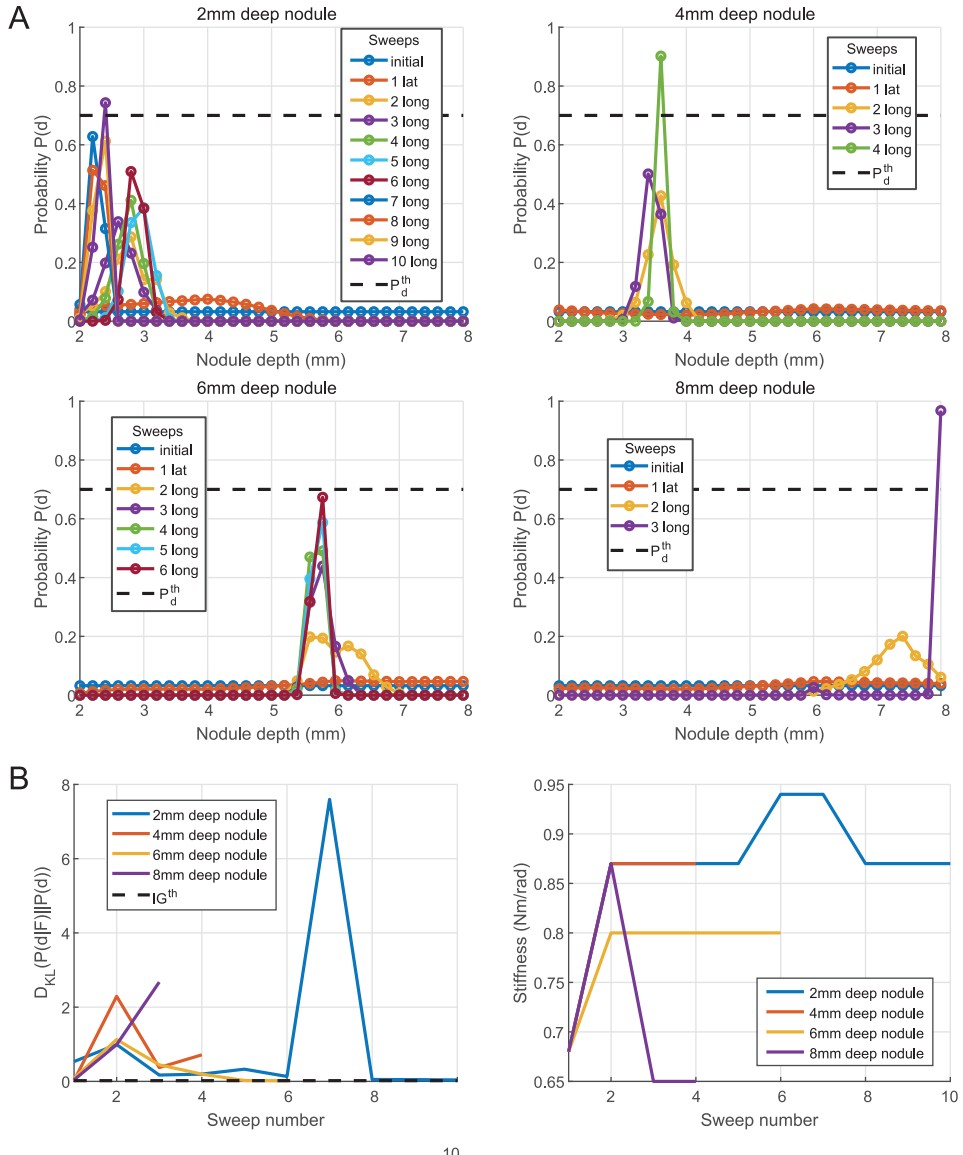

**Fig 14. Algorithm results based on experimental likelihood functions.** (A) Nodule depth estimation by the algorithm. (B) KL divergence and stiffness changes across sweeps.

maximum of 10. The supplemental S4 Video also shows the behavior of the algorithm for a trial over a 4mm deep nodule.

The two thresholds $P_d^{th}$ and $IG^{th}$ can seem redundant at first but they are complementary to avoid too many sweeps. For instance, the majority of the trials are stopped by the threshold $P_d^{th}$, but for the 6mm deep nodule in Fig 14A, the evolution of $P(d)$ over the sweeps is slow, and may not reach the threshold $P_d^{th}$. In this case, $IG^{th}$ allows the algorithm to stop and give a result with lower confidence.

With the proposed threshold values, the algorithm was 100% accurate on the estimation of the presence of a nodule with a single lateral sweep. When a nodule was present, $P(N)$ was over 0.98 after the first sweep, independently from the initial distance between the probe and the nodule. When there was no nodule, P(N) was in a range between 0 and 0.18 after the first

lateral sweep. A trade-off needs to be found while tuning $P_N^{th}$ in order to maximize the detection rate but not increasing the number of false-positive detection, which would lead to unnecessary additional lateral explorations.

Fig 14B shows the evolution of the KL divergence of the nodule depth estimation and the variation of stiffness for the different sweeps for the same trials as shown in Fig 14A. It can be seen that with the set of thresholds, the KL divergence does not need to converge to stop the exploration. Concerning the stiffness, all the trials start with the same stiffness $K_\theta^{detect}$, while the first longitudinal sweep is usually performed with the same stiffness $K_\theta = 0.87$Nm/rad. It is only after the second sweep when $P(d)$ changes significantly that the stiffness selection for the sweeps differs from one nodule depth to another.

On the other hand, Fig 15 shows the results of the algorithm for 3D localization of nodules using the likelihood functions obtained thanks to the FEM simulation instead of the one obtained experimentally. In particular, Fig 15A presents examples of nodule depth estimation for the same four nodules tested previously. One can see that the algorithm is able to detect the depth of the nodule but generally need more sweeps. Thus, the results obtained with the FEM based likelihood functions are particularly good for the 2 and 8mm deep nodules, but for the 4 and 6mm deep nodules the expected accuracy (mean of $P(d)$) and the confidence level (sharpness of $P(d)$) are lower than for the sweeps performed with the likelihood obtained experimentally. This is because the force peak prominence measured during the sweeps is further from the simulated expected value, so the information gained per sweep is generally smaller. As shown in Fig 15C, the threshold on the information gain is reached for the 4 and 6mm deep nodules. Moreover, this figure shows, that the probe can gain information within the first two sweeps with the exception that the 4mm deep nodule requires four sweeps to achieve significant information gain. It can be noticed that this important information gain corresponds to a significant change on the posterior distributions in Fig 15A from a relatively flat distribution to a sharper one and also to a variation of the probe stiffness. This shows once again the importance of stiffness variation for haptic 3D localization of nodules in soft tissues.

The final depth estimate $d_{est}$ can be computed as follows:

$$d_{est} = \sum_{d \in \mathcal{D}} P(d)d \tag{5}$$

The Root Mean Square Error (RMSE) between the estimated depths and the actual nodule depths for all the detections presented in Figs 14 and 15A is 0.27mm. The highest absolute error is 0.53mm for the 6mm deep nodule with the likelihood function obtained from the FE simulation.

In addition, to complement the investigation on the effect of the probe's stiffness variation in the Bayesian nodule depth estimation, 2 trials have been run without the stiffness modulation strategy. During these trials performed for 4 and 6mm nodule depths, the stiffness is therefore maintained constant at the stiffness $K_\theta = 0.68$Nm/rad (the same as the one used for lateral sweeps). The results for these trials are shown in Fig 15B. One can see that the accuracy stays similar to the case where the stiffness is updated from previous knowledge but the confidence level is significantly lower. It can also be observed in Fig 15C that the KL divergence quickly converges to 0, which means that the repetition of the sweeps with the same fixed stiffness did not bring much information on the nodule depth. These results confirm that the proposed strategy using stiffness variation helps in conditioning the force peak prominence likelihood and improves the nodule depth estimation.

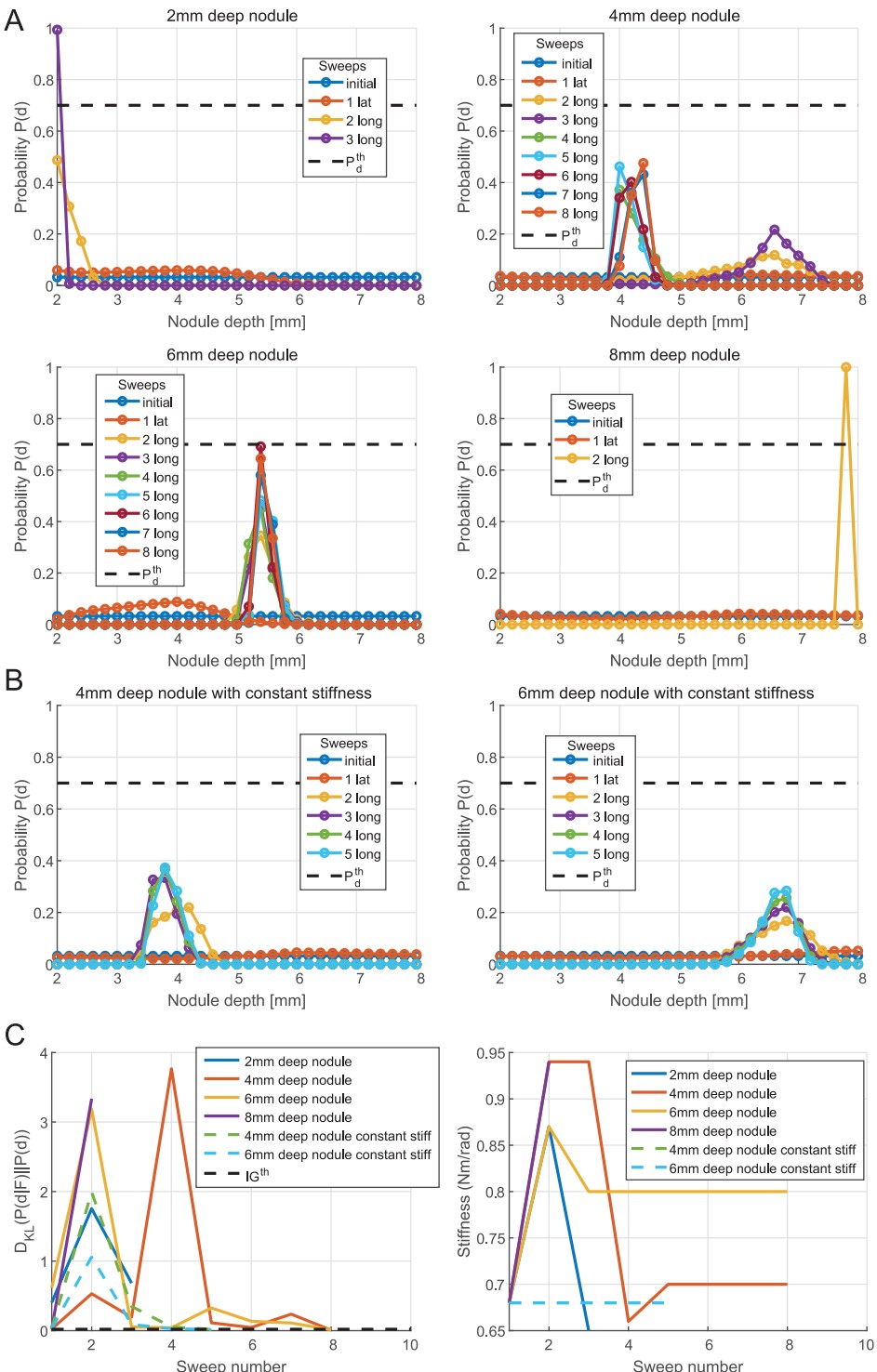

**Fig 15. Algorithm results based on FE simulated likelihood functions.** (A) Nodule depth estimation by the algorithm with the likelihood functions obtained by FEM simulation. (B) Bayesian iterations for nodule depth estimation with a constant stiffness $K_\theta = 0.68$Nm. (C) KL divergence and stiffness changes across sweeps with the likelihood functions obtained by FEM simulation.

Finally, since for all the trials presented in Fig 15, the likelihood functions for lateral sweep used are still the ones obtained from the experiments, there is no significant change in the detection of the nodule's presence.

## Discussion

In this study, we highlighted the importance of tuning the stiffness of a compliant palpation probe to shape the force peak prominence likelihood during a palpation task, and we proposed a controller that utilizes this principle to estimate the 3D localization of a nodule. This study highlights the role of compliance of a soft robot not only as a design parameter for safety but also as a control parameter for improved haptic perception [54].

There is still some remaining work to be done to implement this probing method on real scenarios like automated examination of biopsy samples or even remote patients. The main difficulty is that the likelihood functions obtained in this paper are valid only for the same thickness of soft tissue, the same size of the nodule, and the same sweeping speed since all these factors are impacting the force peak prominence dispersion. To solve this issue, new models able to predict the kinesthetic force uncertainties during a compliant palpation exploration would be needed. We have shown in this paper that FE simulation is one solution to predict the behavior and to generate likelihood functions. Indeed they can be used to generate the likelihood functions when no palpation has been performed on this type of tissue. However, they still require the knowledge of some tissue meta parameters such as the soft tissue elasticity. Also, increasing the accuracy of the tissue model or simulating the lateral sweeps would require developing a 3D FE model and repeating the simulation for several shifts (position along **x**). These modifications would increase the computational cost of the simulation significantly.

Another approach to generate the likelihood functions for different conditions is using mixture models [55, 56]. A mixture model linearly combines a set of nonlinear kernels to fit a given distribution of observations. In this particular scenario, likelihood functions for new materials can be constructed by learning a set of parameters of the mixture model already identified for a known tissue.

In this paper, the flat phantom has been designed to simplify the study and minimize assumptions. In real scenarios, such as breast or abdominal palpations, the surface of the tissue to be palpated is not likely to be flat and the breathing of the patient would add some disturbances. However, these issues have been widely addressed in the literature. Some control strategies can compensate in real time the motion due to the patient breathing [57] or to follow complex and moving surfaces [58–60]. In the future, we will implement these solutions and the Bayesian controller to test the probe on a soft robotic patient phantom with controllable and sensing organs.

More generally, this paper concludes that the probe stiffness matters in conditioning the shape of the likelihood function (sensor model) in a Bayesian framework to estimate a given feature in the tissue (in this case the nodule depth after having identified the location using lateral sweeps). However, when the tissue is inhomogeneous, multiple force prominence values will be present in the force profile. Then a suitable technique should be adopted to filter a target force prominence shape. One realtime solution is to convolve a target force shape on the measured force data. A data-driven approach can be used to build the target shape from a variety of tissue samples with a given feature in them.

In the proposed algorithm, the tactile sensor is currently used to help to find the location of the nodule. Removing the sensor would be possible, but it would require to increase the longitudinal sweeping distance by two times the length of the probe's tip link. This would then

increase the exploration time and also the tissue region area that is probed. These two factors are not really suitable in the case of patient palpation.

Finally, one may notice that the stiffness variations (for the same nodule depth) during the trials with likelihoods functions computed from experimental do not necessary follow the same gradient as the variations during the trials with likelihoods function computed from the FE simulations. More generally, even across trials repeated for the same scenario, we observed different stiffness variations. This comes from the fact that the stiffness is adapted according to the current knowledge during the palpation exploration. The proposed method then presents a strategy that tunes the stiffness in order to increase the probability of getting new information based on prior knowledge (likelihood functions). Due to the stochastic behavior of the force peak prominence, even for the same stiffness and same nodule, the information obtained during a sweep is different from one sweep to another. As a consequence, for two trials on the same nodule, if the information collected so far is different, the stiffness selected to maximize the information gain of the next sweep will be different. Finally, this can also be supported by another study [14] where the variation of the human's joints stiffness during palpation tasks has been studied and clearly showed that this joint stiffness follows a random walk. In other words, during palpation exploration tasks, even humans do not follow a predictable gradient of joint's stiffness variation.

## Conclusion

In this paper, we have shown that the embodied stiffness of the robotic probe is conditioning the force peak prominence likelihood. In particular, we have shown that by controlling the stiffness of the probe it is possible to sharpen the probability distribution of the force peak prominence. This change of shape of the likelihood is seen as a haptic information gain and can be observed for both sweeping directions. For the lateral sweep, the impact of the joint's stiffness variation on the force peak prominence distribution is not as significant as for the longitudinal sweeps. This results in the fact that the haptic information gain is not sufficient to distinguish the depth of a nodule. However, the stiffness can be chosen to facilitate the detection of a nodule. In contrast, for the longitudinal sweeps, the haptic information gain from one depth to another is significant and helps to determine which stiffness is suitable for the depth estimation. To illustrate the importance of conditioning the likelihood of the force peak prominence, we proposed an algorithm that autonomously explores soft tissues using tactile for the localization of the nodule and kinesthetic sensing to estimate the nodule location and depth. This Bayesian algorithm selects the suitable stiffness by minimizing the variance of the expected force prominence based on prior knowledge that can be constructed from past palpations or generated from FE simulations. The algorithm has been tested on a soft tissue phantom with the VLM probe and shows a 100% reliability on the nodules detection and sub-millimeter accuracy in the 3D nodule localization.

## Supporting information

**S1 Appendix. Statistical analysis.** This appendix provides the details of the statistical analysis performed to compare the distribution of the force peak prominence for different stiffnesses. (PDF)

**S1 Fig. FE simulation results.** This supporting figure shows some results obtained during the 2D FE simulation used to model the VLM probe performing a longitudinal sweep on phantoms with a 2mm, with an 8mm nodules, and without nodule respectively. This figure shows the variation in the vertical force and the stress of the phantom (denoted $F_z$ and $\sigma$ respectively)

at two different instants of the sweep, depending on the depth of the nodule and the stiffness.
(EPS)

**S1 Video. Lateral sweeps on a 2mm deep nodule.** This video shows how the lateral sweeps have been performed over a 2mm deep nodule for 3 different shifts and 3 different stiffnesses. (MP4)

**S2 Video. Longitudinal sweeps.** This video shows how the longitudinal sweeps have been performed for 3 different stiffnesses. The video illustrates the sweeps over a 2mm deep nodule and a region of the phantom without nodule.
(MP4)

**S3 Video. Simulation results.** This video shows some results obtained with the FE simulation. 3 different phantom conditions are compared for 2 different stiffnesses.
(MP4)

**S4 Video. Algorithm test with a 4mm deep.** This video shows how the proposed Bayesian algorithm performs over a 4mm deep nodule.
(MP4)

## Author Contributions

**Conceptualization:** Nicolas Herzig, Liang He, Perla Maiolino, Sara-Adela Abad, Thrishantha Nanayakkara.

**Data curation:** Nicolas Herzig.

**Formal analysis:** Nicolas Herzig, Thrishantha Nanayakkara.

**Funding acquisition:** Thrishantha Nanayakkara.

**Investigation:** Nicolas Herzig.

**Methodology:** Nicolas Herzig.

**Project administration:** Nicolas Herzig, Thrishantha Nanayakkara.

**Resources:** Nicolas Herzig, Liang He, Perla Maiolino.

**Software:** Nicolas Herzig, Perla Maiolino.

**Supervision:** Nicolas Herzig, Thrishantha Nanayakkara.

**Validation:** Nicolas Herzig.

**Visualization:** Nicolas Herzig, Sara-Adela Abad.

**Writing – original draft:** Nicolas Herzig, Liang He, Perla Maiolino, Sara-Adela Abad, Thrishantha Nanayakkara.

**Writing – review & editing:** Nicolas Herzig, Liang He, Perla Maiolino, Sara-Adela Abad, Thrishantha Nanayakkara.

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
