## [Decision Letter · Decision Letter 0]

23 Mar 2020

PONE-D-20-03370

Conditioned haptic perception for 3D localization of nodules in soft tissue palpation with a variable stiffness probe

PLOS ONE

Dear Dr. Herzig,

Thank you for submitting your manuscript to PLOS ONE. After careful consideration, we feel that it has merit but does not fully meet PLOS ONE’s publication criteria as it currently stands. Therefore, we invite you to submit a revised version of the manuscript that addresses the points raised during the review process.

The paper is of interest and addresses the important issue of robotic detection and localization of hard nodules in soft tissues. However, all reviewers agreed that the paper need improvents in terms of rigor of experiments and clarity of presentation. In addition, Reviewer 3 raises important points related to the use of a variable stiffness mechanism for this specific application and some discrepancies between conclusions derived from the experiments versus the ones from FEM.

We would appreciate receiving your revised manuscript by May 07 2020 11:59PM. To enhance the reproducibility of your results, we recommend that if applicable you deposit your laboratory protocols in protocols.io, where a protocol can be assigned its own identifier (DOI) such that it can be cited independently in the future. For instructions see: http://journals.plos.org/plosone/s/submission-guidelines#loc-laboratory-protocols

We look forward to receiving your revised manuscript.

Kind regards,

Tommaso Ranzani, PhD

Academic Editor

PLOS ONE

Journal Requirements:

Reviewers' comments:

Reviewer's Responses to Questions

**Comments to the Author**

1. Is the manuscript technically sound, and do the data support the conclusions?

Reviewer #1: Yes

Reviewer #2: Yes

Reviewer #3: Partly

2. Has the statistical analysis been performed appropriately and rigorously? 

Reviewer #1: N/A

Reviewer #2: Yes

Reviewer #3: Yes

3. Have the authors made all data underlying the findings in their manuscript fully available?

Reviewer #1: Yes

Reviewer #2: Yes

Reviewer #3: Yes

4. Is the manuscript presented in an intelligible fashion and written in standard English?

Reviewer #1: Yes

Reviewer #2: Yes

Reviewer #3: Yes

5. Review Comments to the Author

Reviewer #1: Line 5: Authors reported "This trend is opening up new opportunities for robotic applications in the healthcare field." So, it is implied that the healthcare field with robotics is all about tactile feedback, or all the applications include tactile feedback, which is not true at all. It is true that with the tactile feedback, the robotic implications in healthcare field can be improved significantly - but authors should change their original sentence.

Line 37: what do the authors mean by " force and tactile" modalities? Kinesthetic and haptic? Please elaborate or clear these modalities.

Line 147: The previous sentence cites 4 different studies as the previous studies. Then, the authors start talking about presumably one of them saying "In the previous study ……" without specifying which study they are talking about. It should be clarified.

The VLM probe explains the design of the probe, which can be supported also visually. The directions and the definitions should be depicted clearly for the reader.

The pdf version I received had the actual Figure and the figure captions separated. I am not sure if this was a draft problem, or the final manuscript will be like this. If it's the later, numbering the sections make no sense, because the reader cannot follow. The authors should find a way to label the system parts on the images directly. Also, using the arrows with color code based on the motion direction might be impossible for the reader to capture, if they are reading from a black/white copy. Such an identification should be handled differently.

Why exactly Figure 1 (a) and (b) have different coordinate systems? I understand in both conditions, the sweeping action takes place in the y direction but what does that mean? What is the advantage of such rotation for the designer?

The motivation of having two conditions in Figure 1(a) and (b) are not clear. So, the direction of sweeping are different, but they are still tangential to the surface. What is the hypothesis or the expected outcome here? To have different "depth estimation", changing the direction of tangential sweep might be not enough. Also, given Figures have different orientations of component (6), resulting different contact areas with the surface. Is it intentional? If so, how is it related to the sweep direction? If not, why is it different?

Figure 2 is impossible to be understood - possibly because the coordinate system for both conditions are different. Still, this Figure must me improved!

The probe position seems to be changing between trials in the lateral sweep, but not in the longitudinal sweep. Why?

15 different stiffness values have been chosen for the experiment and these values seem random. It seems like these values are changing incrementally, but not linearly (there are some missing values) but it is curious how they are chosen! Is there a reason why all the values between 0.65 and 0.71 was tried, but 0.72 is ignored? Why is the differences between the last4 values are much bigger than the first 4?

FE model is only used for the longitudinal sweep but not for the lateral. Why?

Line 588 : Authors say "This study highlights the role of compliance of a soft robot not only as a design parameter for safety, but also as a control parameter for improved haptic perception " but in the paper, what we see is the comparison between different sweep directions. If these two things are connected, it means authors didn't do a very good job explaining how the sweep direction is related to the probe compliance. It would be also nice to mention this relationship in the discussion and/or conclusion section.

Citation numbers cannot be used as a subject of a sentence ([35] proposed ……)

Reviewer #2: This paper presents a control algorithm for variable lever mechanism probe to detect and localize embedded nodules in soft tissues. Using this algorithm, the 3D position of the nodule can be estimated. In general, this paper is well-written. There are some questions and possible improves below.

1. What is the material used to make the simulated nodules? What types of soft tissues and nodules do you simulate?

2. How do you detect when the contact between the probe and the phantom starts? How does the accuracy of detection of the moments of the contact between the probe and the phantom affect the stiffness estimation?

Reviewer #3: This paper addresses the problem of detecting hard nodules in soft tissue via robotic palpation, as well as assessing their depth. The central ideas are one, that the stiffness of the probe should matter; and two, that the proper stiffness can be found via a Bayesian search technique. The paper is clearly written and technically sound; however, I see very little evidence that supports the central ideas.

First, a few more details: the variable stiffness probe is mounted on a force sensor and equipped with a tactile sensor. Although Fig 7 shows that the latter provides some useful information, as far as I can tell, it is not used to support any of the paper's main points. Therefore, I see the tactile sensor as a bit of a distraction. I would recommend removing it altogether. In any event, the force sensor appears to give a clear indication of nodule location as well as depth. There is no doubt that the robotic probe succeeds!

My concern, however, relates to the importance of probe stiffness.

Figures 6, 9 and 11 illustrate the dependence of the "force peak prominence" on probe stiffness under lateral and longitudinal swiping, in simulation and experiment. With the exception of Fig 11, we see very little dependence on probe stiffness. Even with Fig 11, it would appear to suffice to pick a good stiffness (lower values appear better) and to fix it. The added value of varying the stiffness is by no means apparent.

This brings us to the Bayesian search, in which stiffness was updated trial-over-trial in a Bayesian fashion using likelihood functions at different stiffness values with prominence and nodule depth as variables. Two sets of likelihood functions, one obtained experimentally and one obtained from FEM. In both cases, the Bayesian technique clearly shows the best stiffness varying trial-over-trial. That variation, however, is not good evidence that an optimal stiffness is being obtained. To the contrary, it is notable that the best stiffness versus module depth behaves completely differently in Fig 14 (experimental likelihood) versus Fig 15 (FEM). The former tends toward a softer probe for deeper nodules and the latter tends toward a stiffer probe for deeper nodules. It is deeply concerning that the answers are just the opposite of one another. However, when looking at Fig 13, it appears that the likelihood functions simply don't vary much with stiffness. I suspect that the results are just noise.

Minor point: could the oscillations seen in Figs 8 and 10 be, in part, due to probe dynamics? Perhaps stick-slip excites those dynamics.

6. PLOS authors have the option to publish the peer review history of their article (what does this mean?). If published, this will include your full peer review and any attached files.

Reviewer #1: No

Reviewer #2: Yes: Min Li

Reviewer #3: No

---

## [Author Response · Author response to Decision Letter 0]

14 May 2020

Response to the Reviewer 1

1. Line 5: Authors reported ”This trend is opening up new opportunities for robotic applications in the

healthcare field.” So, it is implied that the healthcare field with robotics is all about tactile feedback,

or all the applications include tactile feedback, which is not true at all. It is true that with the tactile

feedback, the robotic implications in healthcare field can be improved significantly - but authors should

change their original sentence.

The authors would like to thank the reviewer for this comment. The authors did not mean that it is

the only approach that contributes to healthcare robotics. To avoid confusion, the authors rephrased the

sentence as follows:

In section Introduction, paragraph 1 ”This trend is one of the promising advances that can bring new

opportunities for robotic applications in the healthcare field.”

2. Line 37: what do the authors mean by ” force and tactile” modalities? Kinesthetic and haptic? Please

elaborate or clear these modalities.

The authors effectively were meaning kinesthetic instead of force. This has been replaced in the highlighted

sentence but also in the rest of the paper. In order to clarify this point, the authors have also added

these definitions:

In section Related work, paragraph 2 ”The main types of sensors used to do so are the kinesthetic sensors

and tactile sensors. In this paper, we refer to kinesthetic sensors, the sensors that aim to give a signal

related to the force or the torque applied at a probe joint level. On the other hand, the tactile sensing

is referring to fingertip contact sensing using taxel images. Tactile sensing is usually representing the

behavior of the mechanoreceptor at the skin level.”

3. Line 147: The previous sentence cites 4 different studies as the previous studies. Then, the authors start

talking about presumably one of them saying ”In the previous study . . . . . . ” without specifying which

study they are talking about. It should be clarified.

As suggested by the reviewer, the authors are now specifying the study that they were talking about:

In section Related work, paragraph 9 ”In the previous study [47], we have shown that the stiffness of the

arm and hand joints is modified during the longitudinal sweeping exploration of soft tissues by varying

the level of co-contraction of antagonistic muscles.”

4. The VLM probe explains the design of the probe, which can be supported also visually. The directions

and the definitions should be depicted clearly for the reader.

If the authors understood well the comment, the reviewer is asking for a figure describing clearly the

design of the probe and the 2 sweeping directions. The authors have modified Figures 1 and 2. The

modified Fig 1 describes the VLM probe design while the modified Fig 2 clearly depicts the 2 sweeping

directions. We also revised the definitions in the Materials and methods section to clarify the description

of the sweeping directions. The modifications regarding the sweeping directions are detailed in the

comment 7. The modifications regarding the design are the following:

In section Variable stiffness palpation: the VLM probe, paragraph 2 ”Fig 1 shows the design of the VLM

probe. The VLM probe is based on a revolute variable stiffness joint composed of 2 rigid links (the base

link and the tip link) connected with a revolute joint in parallel with a deformable carbon rod. This

carbon rod acts as a variable spring that allows the stiffness of the joint to be controlled thanks to an

Actuonix L12-30-50-6-I linear actuator. This actuator slides the carbon rod through the base link and

the tip link changing the length of the carbon rod that can be bent (active length). As one can see, the hole

in the base link has been designed such as that the carbon rod can slide axially but is constrained radially

to prevent bending of the rod in the base link. On the other hand, the hole in the tip link is large enough

to allow the carbon rod to bend in. A PTFE cylinder is used to transmit the radial forces between the tip

link to the carbon rod. This PTFE cylinder has been designed to slide easily axially when the actuator

is translating the carbon rod. Adjusting the active length of the carbon rod changes, by cantilever effect,

the amount of force required to bend the rod and by consequence the angular stiffness of the probe.”

In section Variable stiffness palpation: the VLM probe, paragraph 3 ”In order to describe the movement

of the probe, we need to define a frame. First, we define the axis z as the direction of the normal to the

phantom surface. We then define the x axis as the intersection between the tangent surface of the phantom

and the mid-sagittal plane of the probe. Finally, the y is defined in order to obtain a direct orthonormal

frame (x,y,z). In the rest of the paper, this reference frame will be used to describe the directions of forces

or displacement.”

5. The pdf version I received had the actual Figure and the figure captions separated. I am not sure if this

was a draft problem, or the final manuscript will be like this. If it’s the later, numbering the sections

make no sense, because the reader cannot follow. The authors should find a way to label the system parts

on the images directly. Also, using the arrows with color code based on the motion direction might be

impossible for the reader to capture, if they are reading from a black/white copy. Such an identification

should be handled differently.

The figures were separated from the captions as explicitly defined in the PLOS submission guideline, the

final manuscript once edited should integrate the figures in the text. However, in order to simplify the

review process, the figures with section numbers have been edited. Also, the caption referring to colors

have been changed to help the reading from grayscale copies.

6. Why exactly Figure 1 (a) and (b) have different coordinate systems? I understand in both conditions, the

sweeping action takes place in the y direction but what does that mean? What is the advantage of such

rotation for the designer?

The difference in the coordinate system was a mistake. Figures 1 and 2 have been modified to avoid

confusion with the reference frame. More details about the motivation of having the two sweeping

directions are given in the next answer.

7. The motivation of having two conditions in Figure 1(a) and (b) are not clear. So, the direction of sweeping

are different, but they are still tangential to the surface. What is the hypothesis or the expected outcome

here? To have different ”depth estimation”, changing the direction of tangential sweep might be not

enough.

The authors thank the reviewer for this comment. The aim of having two different sweeping strategies

is to reproduce the exploration behavior of human participants observed during the previous study [44].

Indeed we observed that different strategies could be applied to localize the nodule and to estimate the

depth. In particular, this study showed that the force applied during the palpation varies according to

the aim of the exploration. Based on these observations and results we found interesting to compare the

two type of sweeping strategies, one local with a light force applied to the phantom and one more global

(with the whole palmar region of the probe’s finger). The hypothesis behind the interest of the sweeping

direction is that one direction is more suitable for nodule localization in the tangential plane where the

other one gives better results to estimate the nodule depth. This hypothesis has been verified through the

study. The authors agree with the fact that changing the direction of the tangential sweep is not the only

impacting factor for depth estimation. The presented paper discusses the interest of stiffness variation as

well. To clarify the information regarding the two sweeping directions, the following modification has

been added to the paper:

In section Sweeping directions, paragraph 2 ”The aim of the two sweeping directions is to reproduce

some human participants’ palpation strategies that we observed during our previous study [44]. We

have shown in this study that the palpation behavior of the participants is adapted to localize the nodule

or to estimate the depth. From these observations and results, we found interesting to compare two types

of sweeping strategies, one local with a light force applied to the phantom using the tip of the probe and

one more global using the whole palmar region of the probe with the tactile sensor.”

8. Also, given Figures have different orientations of component (6), resulting different contact areas with

the surface. Is it intentional? If so, how is it related to the sweep direction? If not, why is it different?

The authors thank the reviewer for this comment. On figure 1, the orientation of the phantom and nodule

(previously labelled (6)) was, effectively, different between the subfigure (a) and (b). The proposed

probing strategy aims to detect the 3D location of the nodules independently from the orientation of the

phantom. This is also the reason why the phantom was also examined in different orientations during

the evaluation of the algorithm. In this regard, the two sweeping directions have been tested for different

orientations of the phantom. The modification of Figures 1 and 2 (presented in our answer to comment 4

of the reviewer 1) should suppress the confusion. To clarify that the strategy aims to be independent of

the initial phantom orientation, the following sentence has been added in the paper:

In section Evaluation of the algorithm, paragraph 1 ”Finally, the proposed palpation strategy aims to

localize the nodule independently from the phantom orientation, so the algorithm has been tested for

several orientations of the phantom.”

9. Figure 2 is impossible to be understood - possibly because the coordinate system for both conditions are

different. Still, this Figure must me improved!

Fig 2 has been modified to improve clarity. The latter is presented in our answer for your comment 4.

10. The probe position seems to be changing between trials in the lateral sweep, but not in the longitudinal

sweep. Why?

As explained in previous answers, the two sweeping directions have different objectives. The interest of

the lateral sweep is to detect the position of the nodule in the tangential plane. To localize the nodule in

the (x,y) plane, the probe utilizes both the kinesthetic sensor and the tactile sensor. Then, once a nodule

is localized, the longitudinal sweeping is used to improve the depth estimation thanks to the kinesthetic

feedback. In this regard, it is interesting to change the position for the lateral sweep (nodule position

detection) but not for the longitudinal sweep (nodule depth estimation). The following paragraph has

been rephrased to improve the clarity of the paper.

In section Sweeping directions, paragraph 6 ”This cycle is repeated 5 times, and after the fifth time, the

VLM probe is shifted by 5mm along x axis to a new initial position. As the lateral sweeps are performed

to localize nodule on a wide area, the aim of this shift is to observe the behavior of the probe when the

latter is sweeping over a nodule at different distances. The next cycle is also repeated 5 times before

applying a new shift. In total, 4 shifts are applied, the distance between the initial and last trajectories is

then 20mm.”

11. 15 different stiffness values have been chosen for the experiment and these values seem random. It

seems like these values are changing incrementally, but not linearly (there are some missing values) but

it is curious how they are chosen! Is there a reason why all the values between 0.65 and 0.71 was tried,

but 0.72 is ignored? Why is the differences between the last 4 values are much bigger than the first 4?

The nonlinearity in the stiffness is coming from the VLM probe behavior. Indeed in a previous study

[22], the VLM probe stiffness has been modeled and characterized for several active lengths of the

carbon rod. Instead of using constant steps in stiffness, the authors have chosen linear steps of active

length of the carbon rod for three reasons: 1) we are using the same active carbon rod length as the one

we characterized in our previous study, 2) it is practically easier to control accurately the position of the

actuator thanks to its sensor (closed-loop control) 3) It is simpler to implement carbon rod displacement

in the FEM simulation. To clarify this information, the following paragraph has been added.

In section Sweeping directions, paragraph 9 ”One can notice that the steps of the stiffness tested in this

paper is not linear. This comes from the fact that for simplicity, we have chosen linear steps of 2mm in

the active length of the carbon rod. This choice allows us to take advantage of the probe characterization

performed in our previous study [22] and makes the stiffness control easier by relying on the closed-loop

position control of the linear actuator. It also simplifies the implementation of carbon rod displacement

in the FE simulation. However, since the relation between the stiffness and the active length of the carbon

rod is nonlinear, it results in nonlinear steps of stiffness.”

12. FE model is only used for the longitudinal sweep but not for the lateral. Why?

The reasons why the FE model for the lateral sweeps is not proposed in the paper are multiple. First,

the aim of our FE model is to provide a further study on the impact of the joint stiffness variation during

palpation exploration. However, the experimental results show that the stiffness variation for the lateral

sweeps is less significant than for the longitudinal sweeps. As a consequence, there was less interest

to further carry the FE simulation for the lateral sweep. Second, to model the lateral sweeps, the FE

simulation needs to be modified from a 2D model to a 3D model. This implies an exponential increase

in computation cost (27 hours were already required to compute the simulation in 2D). Also, to follow

the experimental protocol, the simulation should be performed for the five different shifts, which also

increase the computational cost of the study. Finally, in the proposed algorithm, the lateral sweep is

generally used only once. According to the author, the complexity is not worth the information that the

lateral sweep FEM simulation would bring to this paper. To clarify, the following modifications have

been added to the paper:

In section Finite Element Simulation, paragraph 1 ”The aim of our Finite Element (FE) model is to

provide a further study on the impact of the joint stiffness variation during palpation exploration. The

experimental results show that the variation of stiffness is more significant for the longitudinal sweeps

than for the lateral sweep. As a consequence, we focused our FE simulation on longitudinal sweeps.”

In section Discussion, paragraph 2 ”Also, simulating the lateral sweeps would require developing a 3D

FE model and repeating the simulation for several shifts (position along x). These modifications would

increase the computational cost of the simulation significantly.”

13. Line 588 : Authors say ”This study highlights the role of compliance of a soft robot not only as a design

parameter for safety, but also as a control parameter for improved haptic perception ” but in the paper,

what we see is the comparison between different sweep directions. If these two things are connected,

it means authors didn’t do a very good job explaining how the sweep direction is related to the probe

compliance. It would be also nice to mention this relationship in the discussion and/or conclusion section.

The paper presents the analysis of the impact of the probe stiffness (or compliance) variation on the haptic

(kinesthetic and tactile) detection of nodules in soft tissues for two sweeping directions. The authors

would like to apologize if the confusion comes from the use of the word compliance. Indeed, compliance

is defined as the inverse of the stiffness. The authors then used from time to time compliance instead of

stiffness to avoid repetition. To avoid confusion, the authors have added the following modification in

the paper:

In section Introduction, paragraph 3 ”Since the compliance is defined as the inverse of the stiffness, we

mean by compliant system a physical system with low stiffness. By opposition, a stiff system is a system

with low compliance. We will deliberately use the two words compliance and stiffness inconsistently in

this paper since some ideas are more intuitive when expressed using the stiffness, while some others are

more intuitive with the compliance.”

14. Citation numbers cannot be used as a subject of a sentence ([35] proposed . . . . . . )

As suggested, this has been corrected in the revised manuscript.

In section Related work, paragraph 7 ”Using point by point strategy, Hoshi et al. [36] proposed an

algorithm to optimize the stiffness estimation of the palpated tissues by coupling the force measurement

with a predictive model based on the Finite Element Method”

Response to the Reviewer 2

This paper presents a control algorithm for a variable lever mechanism probe to detect and localize embedded

nodules in soft tissues. Using this algorithm, the 3D position of the nodule can be estimated. In general, this

paper is well-written. There are some questions and possible improves below.

1. What is the material used to make the simulated nodules? What types of soft tissues and nodules do you

simulate?

The components used to simulate the nodule are acrylic spheres of 16mm diameter. The size of the

nodule represents the size of a tumor type T1 (< 2cm) for breast or liver cancer. This is, according to

the TNM classification, the earliest stage where the nodules can be detected by palpation [49]. With the

platinum-catalyzed silicone (Ecoflex 00-10), the authors aim to simulate general human soft tissues, for

instance, abdominal organs such as liver or human breast. Indeed, this type of silicone is widely used to

mimic the mechanical properties of human tissues during palpation [22] or needle insertion [48]. This

information has been added in the paper

In section Soft tissue phantom with nodules, paragraph 2 ”These materials have been widely used to simulate

human soft tissues mechanical properties. In particular, Ecoflex 0010 has been used in biomedical

simulators to practice abdominal palpation [11] or needle insertion [48]. The size of the nodule represents

the size of a tumor of type T1 (<2cm) for the breast or liver cancers. This is, according to the TNM

classification, the earliest stage where the nodules can be detected by palpation [49].”

2. How do you detect when the contact between the probe and the phantom starts? How does the accuracy

of detection of the moments of the contact between the probe and the phantom affect the stiffness

estimation?

The authors thank the reviewer for this question. The contact between the probe and the phantom is

detected every time the probe changes the palpation area thanks to its kinesthetic sensor. As described in

the paper, the method is based on an indentation (without sweeping) and the detection of force variation.

As shown in Fig 4, from the force sensor readings, it is simple to detect when the probe touches the

phantom. Since each actuator has its own position sensors, the position of the probe during the contact

can be found. The accuracy of the contact point detection is important; related studies have shown that a

variation of indentation can impact the stiffness estimation [47, 25]. The described detection routine and

the use of the force peak prominence (which takes into account the force measured around the peak) aims

to improve the robustness against an indentation error. More discussion on this point has been added in

the paper.

In section Variable stiffness palpation: the VLM probe, paragraph 6 ”To autonomously detect the 0mm

indentation position, the method is based on an indentation (without sweeping) and the detection of

variation in the kinesthetic sensor measurement. This detection strategy aims to improve the robustness

of the nodule detection by improving the accuracy of the indentation measurement. This is particularly

important since related studies have shown that a variation of indentation can impact the nodule depth

estimation[47, 25].”

Response to the Reviewer 3

This paper addresses the problem of detecting hard nodules in soft tissue via robotic palpation, as well as

assessing their depth. The central ideas are one, that the stiffness of the probe should matter; and two, that

the proper stiffness can be found via a Bayesian search technique. The paper is clearly written and technically

sound; however, I see very little evidence that supports the central ideas.

1. First, a few more details: the variable stiffness probe is mounted on a force sensor and equipped with a

tactile sensor. Although Fig 7 shows that the latter provides some useful information, as far as I can tell,

it is not used to support any of the paper’s main points. Therefore, I see the tactile sensor as a bit of a

distraction. I would recommend removing it altogether. In any event, the force sensor appears to give a

clear indication of nodule location as well as depth. There is no doubt that the robotic probe succeeds!

The tactile sensor plays an important role in the nodule localization in the (x,y) plane. Indeed as explained

in the algorithm, the tactile sensor is used to locate the xn position of the nodule. The detection of xn after

a lateral sweep is not possible from the force sensor only. Without the x position computed after the

lateral sweep, the longitudinal sweeping distance would have to be increased by two times the length of

the tip of the probe. Increasing this distance would degrade the performances of the proposed method,

in terms of time (longer region to probe several times implies a longer time to find the nodule). In this

regard, the authors decided to keep the tactile sensor in the paper but added more discussion about it.

In section Discussion, paragraph 5 ”In the proposed algorithm, the tactile sensor is currently used to

help to find the location of the nodule. Removing the sensor would be possible, but it would require to

increase the longitudinal sweeping distance by two times the length of the probe’s tip link. This would

then increase the exploration time and also the tissue region area that is probed. These two factors are

not really suitable in the case of patient palpation.”

2. My concern, however, relates to the importance of probe stiffness. Figures 6, 9 and 11 illustrate the

dependence of the ”force peak prominence” on probe stiffness under lateral and longitudinal swiping,

in simulation and experiment. With the exception of Fig 11, we see very little dependence on probe

stiffness. Even with Fig 11, it would appear to suffice to pick a good stiffness (lower values appear

better) and to fix it. The added value of varying the stiffness is by no means apparent.

The authors thank the reviewer for this comment. To provide a generic method that a user could reproduce

for any controllable stiffness probe, we present a detailed statistical analysis of the significance of

joint stiffness variation on the force peak prominence distribution. This statistical analysis supports the

authors’ claim that the stiffness plays a significant role in the force peak prominence distribution during

longitudinal sweeps and by consequence on the nodule depth estimation. Also, the case where a low

fix stiffness (Kq = 0:68Nm/rad) is maintained across trials have been tested and added to the Fig 15 to

highlight the limitation of such a strategy in term of estimation confidence.

The presented statistical analysis tests the null hypothesis that the data from 2 different stiffnesses are

coming from the same distribution. Since the distribution for each stiffness is not normally distributed,

we use the Kruskal-Wallis test, which is particularly suitable for non-parametric distributions. The results

are summarized in the supplemental Appendix S1. These results of cross Kruskal-Wallis tests

show that, for the longitudinal sweeps with nodules, the force peak prominence is statistically different

(p-value< 0:05). In addition, the more different the two stiffnesses are, the higher the significance of

the difference between their force peak prominence distribution is. Furthermore, the results from the

Kruskal-Wallis tests for data from the lateral sweeps show that the difference between the force peak

prominence’s distributions is not statistically significant. These results support our claim that the stiffness

variation has a lower significance for the lateral sweeps. Finally, the results of the Kruskal-Wallis

tests for the longitudinal sweeps data when no nodule is embedded exhibits a lower number of statistically

different distributions than the ones from the data with a nodule. This can be interpreted as the fact

that the interaction between the nodule and stiffness impacts the force peak prominence significantly. In

order to make clearer that the stiffness variation is more significant for the longitudinal sweeps than for

the lateral sweeps, the authors have added the following paragraphs:

In section Conclusion, paragraph 1 ”For the lateral sweep, the impact of the joint’s stiffness variation on

the force peak prominence distribution is not as significant as for the longitudinal sweeps. This results in

the fact that the haptic information gain is not sufficient to distinguish the depth of a nodule. However, the

stiffness can be chosen to facilitate the detection of a nodule. In contrast, for the longitudinal sweeps, the

haptic information gain from one depth to another is significant and helps to determine which stiffness is

suitable for the depth estimation.”

In section Lateral sweep, paragraph 10 ”To further support the interpretation of Fig 6, we detail, in the

supplemental Appendix S1, a comparison of the distributions obtained for each stiffness using statistical

analysis.”

In section Experimental results, paragraph 2 ”Similarly to the lateral sweeps, the significance of the

probe’s stiffness variation on the force peak prominence distribution is further studied, using statistical

analysis, in the supplemental Appendix S1.”

To highlight the interest of the proposed algorithm compared to a palpation strategy with constant stiffness.

Two examples of Bayesian nodule depth estimation with a constant stiffness (chosen low as recommended

by the reviewer) have been added to the figure 15. These 2 examples clearly show that the

information gain (KL divergence) drops quickly, and the final confidence level of the depth estimation

stays low but with good accuracy. The discussion added to compare the scenario with a constant stiffness

with the proposed algorithm follows:

In section Evaluation of the algorithm, paragraph 8 ”In addition, to complement the investigation on the

effect of the probe’s stiffness variation in the Bayesian nodule depth estimation, 2 trials have been run

without the stiffness modulation strategy. During these trials performed for 4 and 6mm nodule depths,

the stiffness is therefore maintained constant at the stiffness Kq = 0:68Nm/rad (the same as the one used

for lateral sweeps). The results for these trials are shown in Fig 15B. One can see that the accuracy

stays similar to the case where the stiffness is updated from previous knowledge but the confidence level

is significantly lower. It can also be observed in Fig 15C that the KL divergence quickly converges to 0,

which means that the repetition of the sweeps with the same fixed stiffness did not bring much information

on the nodule depth. These results confirm that the proposed strategy using stiffness variation helps in

conditioning the force peak prominence likelihood and improves the nodule depth estimation.”

3. This brings us to the Bayesian search, in which stiffness was updated trial-over-trial in a Bayesian fashion

using likelihood functions at different stiffness values with prominence and nodule depth as variables.

Two sets of likelihood functions, one obtained experimentally and one obtained from FEM. In both cases,

the Bayesian technique clearly shows the best stiffness varying trial-over-trial. That variation, however,

is not good evidence that an optimal stiffness is being obtained. To the contrary, it is notable that the best

stiffness versus module depth behaves completely differently in Fig 14 (experimental likelihood) versus

Fig 15 (FEM). The former tends toward a softer probe for deeper nodules and the latter tends toward a

stiffer probe for deeper nodules. It is deeply concerning that the answers are just the opposite of one

another. However, when looking at Fig 13, it appears that the likelihood functions simply don’t vary

much with stiffness. I suspect that the results are just noise.

The authors thank the reviewer for this comment. This paper shows how a set of favorable likelihood

functions can be used for fast convergence of the posterior to a higher information gain. The paper shows

that this likelihood function conditioning can be done just by changing the stiffness.

The authors did not claim that a global optimal stiffness exists. Indeed, the claim of the author is even

the opposite saying that the stiffness needs to be adapted according to the current knowledge during the

palpation exploration. The proposed method then presents a strategy that tunes the stiffness in order

to increase the probability of getting new information based on prior knowledge (likelihood functions).

Comparing the gradient of the selected stiffness between 2 different trials does not really make sense

since it is dependent on the past measurement of a random variable (the force peak prominence). Due

to the stochastic behavior of this variable, even for the same stiffness and same nodule, the information

obtained is different (which is the purpose of the paper). By consequence, it is not surprising that the

stiffness gradient followed during the trials with the likelihood functions obtained from the FEM (Fig 15)

and the likelihood functions obtained from experimental data (Fig 14) are different. This comes from the

fact that the prior knowledge is different and so is the information remained to be obtained. This can also

be supported by another study [14] where the variation of the human’s joints stiffness during palpation

tasks has been studied and clearly showed that this joint stiffness follows a random walk. This confirms

that during palpation exploration tasks even humans do not follow a particular gradient of joint stiffness

variation.

Even if in Fig 13, the variation of the likelihood function for different stiffness is not visually obvious, the

added statistical analysis clearly shows that the variation of the force peak prominence distribution due

to a stiffness variation is statistically significant for the longitudinal sweeps. The discussions obtained

by addressing the reviewer’s comments have been used to strengthen the discussions in the paper. The

modifications done are the following:

In section Discussion, paragraph 6 ”Finally, one may notice that the stiffness variations (for the same

nodule depth) during the trials with likelihoods functions computed from experimental do not necessary

follow the same gradient as the variations during the trials with likelihoods function computed from

the FE simulations. More generally, even across trials repeated for the same scenario, we observed

different stiffness variations. This comes from the fact that the stiffness is adapted according to the current

knowledge during the palpation exploration. The proposed method then presents a strategy that tunes

the stiffness in order to increase the probability of getting new information based on prior knowledge

(likelihood functions). Due to the stochastic behavior of the force peak prominence, even for the same

stiffness and same nodule, the information obtained during a sweep is different from one sweep to another.

As a consequence, for two trials on the same nodule, if the information collected so far is different, the

stiffness selected to maximize the information gain of the next sweep will be different. Finally, this

can also be supported by another study [14] where the variation of the human’s joints stiffness during

palpation tasks has been studied and clearly showed that this joint stiffness follows a random walk. In

other words, during palpation exploration tasks, even humans do not follow a predictable gradient of

joint’s stiffness variation.”

In section Experimental results, paragraph 5 ”In particular, the bests of the tested stiffnesses to detect

the presence of a nodule in the stiffness range of the VLM probe are respectively Kq = 0:68Nm/rad and

Kq = 0:76Nm/rad for the lateral sweep and for the longitudinal sweep.”to avoid the use of ”optimal”.

4. Minor point: could the oscillations seen in Figs 8 and 10 be, in part, due to probe dynamics? Perhaps

stick-slip excites those dynamics.

These oscillations come from both the dynamics of the probe and the dynamics of the phantom (both

connected in series). Indeed, if they were coming exclusively from the probe dynamics, the oscillations

would vary with the stiffness variation but not with the nodule depth. Also, one can see that for the same

stiffness, the amplitude of the oscillations is varying with the nodule depth (also visible on simulation

from Fig 8); so, both dynamics are important. The authors strongly agree on the fact that the stick-slip of

the probe excites the probe and phantom dynamics. For improving clarity, more discussion about these

oscillations has been added in the paper.

In section Simulation results, paragraph 3 ”Furthermore, one can notice that the amplitude of these

oscillations varies with the nodule depth, which means that these oscillations are not only dependent on

the probe’s internal dynamics but also on the ones from the phantom.”

---

## [Decision Letter · Decision Letter 1]

8 Jun 2020

PONE-D-20-03370R1

Conditioned haptic perception for 3D localization of nodules in soft tissue palpation with a variable stiffness probe.

PLOS ONE

Dear Dr. Herzig,

Thank you for submitting your manuscript to PLOS ONE. After careful consideration, we feel that it has merit but does not fully meet PLOS ONE’s publication criteria as it currently stands. Therefore, we invite you to submit a revised version of the manuscript that addresses the points raised during the review process.

The authors should still address the comments from the reviewer in particular on: (1) how the experimental results can be generalized to non-homogeneous curved objects, (2) the validation of the accuracy of the proposed methodology, and  (3) the modeling approach.

We look forward to receiving your revised manuscript.

Kind regards,

Tommaso Ranzani, PhD

Academic Editor

PLOS ONE

Reviewers' comments:

Reviewer's Responses to Questions

**Comments to the Author**

1. If the authors have adequately addressed your comments raised in a previous round of review and you feel that this manuscript is now acceptable for publication, you may indicate that here to bypass the “Comments to the Author” section, enter your conflict of interest statement in the “Confidential to Editor” section, and submit your "Accept" recommendation.

Reviewer #3: All comments have been addressed

Reviewer #4: (No Response)

2. Is the manuscript technically sound, and do the data support the conclusions?

Reviewer #3: Yes

Reviewer #4: Partly

3. Has the statistical analysis been performed appropriately and rigorously? 

Reviewer #3: Yes

Reviewer #4: Yes

4. Have the authors made all data underlying the findings in their manuscript fully available?

Reviewer #3: Yes

Reviewer #4: Yes

5. Is the manuscript presented in an intelligible fashion and written in standard English?

Reviewer #3: Yes

Reviewer #4: Yes

6. Review Comments to the Author

Reviewer #3: Thank you for the thorough response to my critique and for adding the constant stiffness case as a comparison. I also stand corrected on the behavior of the stiffness during Bayesian search. I now find the paper quite convincing.

Reviewer #4: The authors present a new method for 3D localization of nodules in sot tissues based on a variable stiffness probe equipped with F/T sensors for kinesthetic perception and a tactile array for tactile perception. The authors proposed an exploration strategy based on a Bayesian approach to the detection and localization of the nodule that allows the authors to set the stiffness of the probe and the direction of the sweep to detect and localize the nodule.

The paper is well written and clear. However, I have some doubts and concerns that I would like the authors to clarify.

Major concerns

1) The proposed study is based on the experimental results over a simplified setup in which the soft tissue is isotropic and homogenous, and its surface is flat. Since the method relies on fine-tuning of a few thresholds, how does it generalize to the real case, where the conditions on the soft tissue are not so clean, and the presence of other organs and tissues (even as stiff as bones) affects the sensed force?

2) I don't see a great added value in the FEM simulations that the authors propose. First, I would like to have more details about how contacts were modeled (are there bearings, bushes, ... ?), and about the size of the soft tissue and its constraints to a fixed frame Second, if it can have a role in setting the initial value of the stiffness for a real case experiment, the soft tissue should be modeled more accurately.

3) My third concern is about parameters. In particular the soft tissue thickness, the nodule Young modulus, and its depth. Since nodule diameter is 16mm, I would expect to have at least 32mm of soft tissue between the nodule and the supporting rigid plane. Even if the applied forces are small, it is a bit surprising that the nodule, especially at 2mm depth, does not influence the stress tensor of the surrounding soft tissue. For what regarding the depth, why the maximum selected depth is 8mm?

4) The proposed algorithm seems quite sensitive to the thresholds. In all cases one lateral sweep allows the system to make the right decision whether the nodule is present or not. Did the authors try starting from a position far from the nodule? Is it so unlikely to have P(N)<0.2 after the first sweep if it takes place far from the nodule?

5) The authors claim submillimeter accuracy, but this is not supported by the results. Please note that it is even difficult to place the nodule in the soft body with such accuracy. Moreover, I can't see in the results where such accuracy has been achieved.

Minor concerns

6) Algorithm1: I suggest the inclusion of an escape from the do-while (e.g. a timeout). Line 15: shouldn't dir be "lat" instead of "long"? Line 18: shouldn't dir be "long" instead of "lat"?

7) line 221: what does it mean "implemented in real-time"?

8) Typos: line 334: remove "performed", line 462: "sweeps", line 488: "cases", line 512: remove "distance"

7. PLOS authors have the option to publish the peer review history of their article (what does this mean?). If published, this will include your full peer review and any attached files.

Reviewer #3: No

Reviewer #4: No

---

## [Author Response · Author response to Decision Letter 1]

29 Jun 2020

Response to the Reviewer 3

Thank you for the thorough response to my critique and for adding the constant stiffness case as a comparison.

I also stand corrected on the behavior of the stiffness during Bayesian search. I now find the paper quite

convincing.

The authors would like to thank again the reviewer for the valuable comments and feedback that helped us to

improve the clarity of the paper. The authors are happy to read that the previous answers reached the expectation

of the reviewer.

Response to the Reviewer 4

The authors present a new method for 3D localization of nodules in sot tissues based on a variable stiffness

probe equipped with F/T sensors for kinesthetic perception and a tactile array for tactile perception. The authors

proposed an exploration strategy based on a Bayesian approach to the detection and localization of the nodule

that allows the authors to set the stiffness of the probe and the direction of the sweep to detect and localize the

nodule. The paper is well written and clear. However, I have some doubts and concerns that I would like the

authors to clarify.

1. The proposed study is based on the experimental results over a simplified setup in which the soft tissue

is isotropic and homogenous, and its surface is flat. Since the method relies on fine-tuning of a few

thresholds, how does it generalize to the real case, where the conditions on the soft tissue are not so

clean, and the presence of other organs and tissues (even as stiff as bones) affects the sensed force?

The aim of this paper is to quantify the role of compliance in the probe to accurately estimate the depth

of a stiff formation. Therefore, we idealized the scenario as much as possible to remove the effect of

any other artefacts introduced by the tissue. We even use a spherical hard nodule to remove the effect

of uneven geometry of the nodule. We do not claim that there is a specific probe stiffness that gives the

best result irrespective of the probe and the tissue condition. Therefore, what can be generalized is that

probe stiffness matters in conditioning the shape of the likelihood function (sensor model) in a Bayesian

framework to estimate a given feature in the tissue (in this case the nodule depth after having identified the

location using lateral sweeps). Therefore, when the tissue is inhomogeneous, multiple force prominence

values will be present in the force profile. Then a shape filter such as a convolution of a target shape over

the force profile should be developed to extract the target feature. This is beyond the scope of this paper.

We added the following text in the discussion to address this concern:

In section Discussion, paragraph 5 ”More generally, this paper concludes that the probe stiffness matters

in conditioning the shape of the likelihood function (sensor model) in a Bayesian framework to estimate a

given feature in the tissue (in this case the nodule depth after having identified the location using lateral

sweeps). However, when the tissue is inhomogeneous, multiple force prominence values will be present in

the force profile. Then a suitable technique should be adopted to filter a target force prominence shape.

One realtime solution is to convolve a target force shape on the measured force data. A data-driven

approach can be used to build the target shape from a variety of tissue samples with a given feature in

them.”

2. I don’t see a great added value in the FEM simulations that the authors propose. First, I would like to

have more details about how contacts were modeled (are there bearings, bushes, ... ?), and about the size

of the soft tissue and its constraints to a fixed frame Second, if it can have a role in setting the initial value

of the stiffness for a real case experiment, the soft tissue should be modeled more accurately.

The FEM simulations were done to understand how the probe stiffness variation leads to changes in

tissue stress dynamics that cannot be measured otherwise. Such insights are useful to understand why

probe stiffness variation and resulting differences in the shape of likelihood functions are underpinned

by the coupled dynamics between the probe and the tissue. Also, we have shown that the FE Analysis

can be used to generate the likelihood function. Using this likelihood functions, generated from simulations,

comes at a cost on the confidence level of the prediction, but can be useful in the case where

no experimental palpation data is available. Regarding the contact models; unfortunately, COMSOL

documentation does not specify if bearings or bushes models are used. However, the model is defined

with 2 contact pairs: the fingertip and the phantom. Then the pressure contact calculation between the

2 contact pairs is based on an Augmented Lagrangian Method, and the friction between the 2 pairs is

modeled as Coulomb friction. Finally, no rolling resistance is modeled for the contact between the probe

and the phantom, which is equivalent to assuming pure sliding. The authors agree that there is still room

for improvement in the soft tissue model. Yet, for the proposed application, the FE model is accurate

enough to perform the 3D localization of the nodules based on the likelihood functions obtained from the

simulation results. To improve the clarity of the model approach and support the interest of the proposed

FE simulation, the following modifications have been added to the paper:

In section Finite Element Simulation, paragraph 5 ”The contact between the probe and the phantom is

modeled with two surface contact pairs covering the palmar region of the probe and the upper layer of the

phantom, respectively. The pressure contact calculation is based on an Augmented Lagrangian Method,

and the friction between the 2 contacts is modeled as Coulomb friction (m in Table 1). Finally, no rolling

resistance is modeled for the contact between the probe and the phantom assuming pure sliding at the

elements level.”

In section Discussion, paragraph 2 ”Also, increasing the accuracy of the tissue model or simulating

the lateral sweeps would require developing a 3D FE model and repeating the simulation for several

shifts (position along x). These modifications would increase the computational cost of the simulation

significantly.”

3. My third concern is about parameters. In particular the soft tissue thickness, the nodule Young modulus,

and its depth. Since nodule diameter is 16mm, I would expect to have at least 32mm of soft tissue between

the nodule and the supporting rigid plane. Even if the applied forces are small, it is a bit surprising that

the nodule, especially at 2mm depth, does not influence the stress tensor of the surrounding soft tissue.

For what regarding the depth, why the maximum selected depth is 8mm?

The phantom has been designed with a similar ratio tissue thickness to nodule diameter than the ones

used in published related works such as [25, 47, 29]. Also, it can be noticed that in the case of T1 tumors

in the kidney or in the left lobe of the liver, for instance, the ratio between the organ thickness and tumor

thickness can be inferior to 2. As the reviewer noticed, the FE simulation shows that with the small

forces applied during the longitudinal sweeps, the stress tensor under the nodule is not influenced even

for the 2mm deep nodule. This phenomenon comes from the fact that during the palpation sweeps, the

displacement of the nodule is negligible compared to the displacement of tissue above the nodule. In

other words, the amount of tissue compressed between the probe and the nodule (referred in this article

as the nodule depth) is more significant than the amount of tissue under the nodule. Finally, we decided

to limit the nodule depth to 8mm to reduce the average palpation force level to minimise damage to the

probe. Indeed, as shown in related works, detecting deeper nodules requires deeper indentation and by

consequence, higher forces. Higher forces also lead to faster degradation of the tissue in repeated trials,

making it difficult to compare results. Moreover, the lateral sweeps used an array of capacitive tactile

sensors, that saturate if an excessive force is applied during lateral sweeps to locate the nodule. Since

the development of a tactile sensor array for deep tissue exploration is not the focus of this paper, we

used a nodule depth that meets all hardware requirements to demonstrate the key scientific phenomenon

mentioned above. The discussion generated by the reviewer’s comment have been added in the paper as

follows:

In section Soft tissue phantom with nodules, paragraph 1 ”The ratio between the tissue thickness and

the nodule diameter has been chosen accordingly to the one used in related studies in the literature

[25, 47, 29]”

In section Soft tissue phantom with nodules, paragraph 2 ”In the presented study, we limited the nodule

depth to 8mm to reduce the average palpation force level and minimize damage to the probe. Indeed,

as shown in related works, detecting deeper nodules requires deeper indentation and by consequence,

higher forces. Higher forces also lead to faster degradation of the tissue in repeated trials, making it

difficult to compare results. Moreover, the lateral sweeps used an array of capacitive tactile sensors, that

saturates if an excessive force is applied to locate the nodule. To avoid saturating the tactile sensor, we

used a nodule depth that meets all hardware requirements to demonstrate the role of stiffness variation

in conditioning the haptic perception during 3D localization of nodules in soft tissues. ”

In section Simulation results, paragraph 2 ”Moreover, with the small forces applied during the longitudinal

sweeps, the stress in the material under the nodule is not impacted as much as the stress in the

material above the nodule. This phenomenon comes from the fact that during the palpation sweeps, the

displacement of the nodule is small compared to the displacement of tissue above the nodule. This shows

that the probe is more significantly affected by the amount of material above the nodule (the nodule

depth) than the amount of material under the nodule.”

4. The proposed algorithm seems quite sensitive to the thresholds. In all cases one lateral sweep allows the

system to make the right decision whether the nodule is present or not. Did the authors try starting from

a position far from the nodule? Is it so unlikely to have P(N) < 0:2 after the first sweep if it takes place

far from the nodule?

As any control strategy, the proposed algorithm relies on the tuning of some parameters (the threshold

here). In the sake of clarity, the authors described and discuss the role of each chosen threshold. One of

the advantages of the proposed threshold is that it is simple for the user to tune them according to their

desired level of confidence. The authors proposed this threshold because it is the one that gave the best

detection rate during the validation of the algorithm. The authors tested the algorithm at different starting

distance from the nodule, and the detection was successful independently from the starting point. The

limitation of this threshold comes more from the risk of false-positive than from missing a nodule due to

the initial position of the probe. Indeed, the algorithm continues to investigate even if the probability of

having a nodule is higher or equal 20%, which may result in some case in additional lateral exploration

when it was not required. The discussion from this question has been added to the paper:

In section Evaluation of the algorithm, paragraph 4 ”With the proposed threshold values, the algorithm

was 100% accurate on the estimation of the presence of a nodule with a single lateral sweep. When

a nodule was present, P(N) was over 0.98 after the first sweep, independently from the initial distance

between the probe and the nodule. When there was no nodule, P(N) was in a range between 0 and 0.18

after the first lateral sweep. A trade-off needs to be found while tuning Pth

N in order to maximize the detection rate but not increasing the number of false-positive detection, which would lead to unnecessary additional lateral explorations.”

5. The authors claim submillimeter accuracy, but this is not supported by the results. Please note that it is

even difficult to place the nodule in the soft body with such accuracy. Moreover, I can’t see in the results

where such accuracy has been achieved.

The authors disagree on this point with the reviewer. Indeed, the Root Mean Square Error (RMSE) for

the eight nodule detections using the presented algorithm (the two trials with fixed stiffness are removed

since they do not represent the algorithm) is 0:27mm. The highest absolute error is 0:53mm for the

6mm deep nodule with the likelihood function obtained from the FE simulation. Both the RMSE and the

maximum absolute error are under 1mm, which support the authors claim on submillimeter accuracy. The

accuracy of the nodule detection is computed from the estimated depth computed from the probability

distribution obtained after the last sweep (when the algorithm ends) as follow:

dest = sum(P(d)d)

Of course, the computed errors relies on the fact that the position of the nodule tested to generate the

likelihood functions are the ground truth for 2, 4, 6 and 8 mm deep. To better support the author claim,

the following modifications have been added to the paper:

In section Evaluation of the algorithm, paragraph 7 ”The final depth estimate dest can be computed as

follows:

dest = dest = sum(P(d)d)

The Root Mean Square Error (RMSE) between the estimated depths and the actual nodule depths for all

the detections presented in Fig 14 and Fig 15A is 0.27mm. The highest absolute error is 0.53mm for the

6mm deep nodule with the likelihood function obtained from the FE simulation.”

6. Algorithm1: I suggest the inclusion of an escape from the do-while (e.g. a timeout). Line 15: shouldn’t

dir be ”lat” instead of ”long”? Line 18: shouldn’t dir be ”long” instead of ”lat”?

The authors would like to thanks the authors for his suggestion. Adding a timeout to the algorithm would

probably increase the robustness of the algorithm for real-world scenarios. However, for the purpose of

this paper, the author does not feel the need to add one since the threshold on the information gain already

act like a timeout. Indeed when the information gain during a new sweep is too low, the algorithm stops

the sweeps and returns the estimated position and depth probability distribution. Thank you very much

for pointing out the typos for line 15 and 18, the directions have corrected.

7. line 221: what does it mean ”implemented in real-time”?

The authors were using inappropriately ”real-time”, thank you for pointing it out. The sentence has now

been rephrased as follows:

In section Variable stiffness palpation: the VLM probe, paragraph 4 ”The programs to run the experiment

and the algorithm have been implemented using C++.”

8. Typos: line 334: remove ”performed”, line 462: ”sweeps”, line 488: ”cases”, line 512: remove ”distance”

The authors would like to thank the reviewer once again for pointing these typos out. They have been

corrected.

---

## [Decision Letter · Decision Letter 2]

27 Jul 2020

Conditioned haptic perception for 3D localization of nodules in soft tissue palpation with a variable stiffness probe.

PONE-D-20-03370R2

Dear Dr. Herzig,

We’re pleased to inform you that your manuscript has been judged scientifically suitable for publication and will be formally accepted for publication once it meets all outstanding technical requirements.

Kind regards,

Tommaso Ranzani, PhD

Academic Editor

PLOS ONE

Additional Editor Comments (optional):

Reviewers' comments:

Reviewer's Responses to Questions

**Comments to the Author**

1. If the authors have adequately addressed your comments raised in a previous round of review and you feel that this manuscript is now acceptable for publication, you may indicate that here to bypass the “Comments to the Author” section, enter your conflict of interest statement in the “Confidential to Editor” section, and submit your "Accept" recommendation.

Reviewer #4: All comments have been addressed

2. Is the manuscript technically sound, and do the data support the conclusions?

Reviewer #4: Yes

3. Has the statistical analysis been performed appropriately and rigorously? 

Reviewer #4: Yes

4. Have the authors made all data underlying the findings in their manuscript fully available?

Reviewer #4: Yes

5. Is the manuscript presented in an intelligible fashion and written in standard English?

Reviewer #4: Yes

6. Review Comments to the Author

Reviewer #4: Thank you for addressing my concerns, pointing out wha aspects are to be included in the focus of the paper. I believe that the paper can be published in the present form.

7. PLOS authors have the option to publish the peer review history of their article (what does this mean?). If published, this will include your full peer review and any attached files.

Reviewer #4: No

---

## [Editor Report · Acceptance letter]

30 Jul 2020

PONE-D-20-03370R2 

Conditioned haptic perception for 3D localization of nodules in soft tissue palpation with a variable stiffness probe. 

Dear Dr. Herzig:

I'm pleased to inform you that your manuscript has been deemed suitable for publication in PLOS ONE. Congratulations! Your manuscript is now with our production department. 

Kind regards, 

on behalf of

Dr. Tommaso Ranzani 

Academic Editor

PLOS ONE